# Multidecadal Preconditioning of the Maud Rise Polynya Region

René M. van Westen[1] and Henk A. Dijkstra[1,2]

[1]Institute for Marine and Atmospheric research Utrecht, Department of Physics, Utrecht University, Utrecht, the Netherlands
[2]Center for Complex Systems Studies, Utrecht University, Utrecht, the Netherlands

*Correspondence to:* René van Westen <r.m.vanwesten@uu.nl>

**Abstract.** In this paper, we consider Maud Rise polynya formation in a long (250 years) high-resolution (ocean 0.1°, atmosphere 0.5° horizontal model resolution) of the Community Earth System Model. We find a dominant multidecadal time scale in the occurrence of these Maud Rise polynyas. Analysis of the results leads us to the interpretation that a preferred time scale can be induced by the variability of the Weddell Gyre, previously identified as the Southern Ocean Mode. The large-scale pattern of heat content variability associated with the Southern Ocean Mode modifies the stratification in the Maud Rise region and leads to a preferred time scale in convection through preconditioning of the subsurface density, and consequently to polynya formation.

## 1 Introduction

A polynya is an open-water area enclosed by sea ice which persists at least for a few months. A famous area for polynya formation is the Maud Rise region in the Weddell Sea. Polynyas occurring at this location are usually referred to Maud Rise Polynyas (MRPs), referring to the bathymetry feature below the ocean water. In 1974, the first MRP was observed and developed into the larger Weddell Polynya in the following years (Carsey, 1980). These polynyas were observed by in situ (Gordon, 1978) and satellite microwave imaging (Carsey, 1980). The mid-1970s polynya was characterised by a sea-ice enclosed open water area $(1 - 3 \times 10^5$ km$^2$) located near 0°E and 65°S (Gordon, 1978). The interest for polynyas has increased recently because of the appearance of an MRP in the austral winter and spring of 2017, with an open-water area of about $0.5 \times 10^5$ km$^2$ (Campbell et al., 2019).

Many observational, modelling and theoretical studies based on the 1974 – 1976 polynya have addressed its formation, evolution and decay (Martinson et al., 1981; Parkinson, 1983; Holland, 2001; Gordon et al., 2007; Martin et al., 2013; Latif et al., 2017; Weijer et al., 2017; Kurtakoti et al., 2018). The occurrence of convection in the vicinity of Maud Rise is clearly a generic aspect during MRP formation. Convective events are induced by a destabilisation of the water column and the classical view is that surface salinity anomalies (e.g. through brine rejection) induce convection once the column is preconditioned through subsurface processes (Martinson et al., 1981). In Gordon et al. (2007), it is proposed that a negative phase of the Southern Annular Mode (SAM) causes drier atmospheric conditions (net evaporation) resulting in a more saline surface layer leading to a destabilisation of the water column. Preconditioning can also occur due to the interaction of an ocean eddy and the Maud Rise topographic feature (Holland, 2001) and by stratified Taylor columns (Alverson and Owens, 1996; de Steur et al., 2007).

The occurrence of Southern Ocean convection is also ubiquitous in many global climate models (GCMs). Martin et al. (2013) show that in the Kiel Climate Model (KCM) a centennial time scale build-up of a heat reservoir at mid-level depths (1000 – 3000 m) preconditions the water column affecting convection. Reintges et al. (2017) analysed a variety of (Coupled Model Intercomparison Project phase five, CMIP5) GCMs and found that several models display deep convection events at multidecadal timescales, for example the GFDL-CM (Zanowski et al., 2015; Zhang and Delworth, 2016). Reintges et al. (2017) argue that models with a weak (strong) stable stratification tend to have a shorter (longer) re-occurrence time of deep convection, and demonstrate the importance of sea-ice volume on the average length of both non-convective and convective periods.

Dufour et al. (2017) analysed MRP events in two versions (eddy-permitting and eddy-resolving) of the GFDL-CM. The lower resolution version has a weaker vertical background stratification (compared to the high-resolution version) and displays quasi-continuous deep convection events. The effects of eddy-driven heat- and salt transports and the representation of overflows were shown to be crucial for the stratification changes in the Maud Rise region (Dufour et al., 2017). In addition, Weijer et al. (2017) demonstrated in a 100-year model simulation of the Community Earth System Model (CESM) that the MRP is only present in the eddying-version of the same model. The results in Dufour et al. (2017) and Weijer et al. (2017) suggest that ocean eddies (and also overflows) play a significant role in the vertical background stratification (in addition to the effects already mentioned by Reintges et al. (2017)) and therefore affect the time scale of the occurrence of deep convection.

The diverse multidecadal (Reintges et al., 2017) to centennial (Martin et al., 2013) variability found in climate models gives the impression that convection can occur rather randomly through surface density perturbations once the water column is sufficiently preconditioned. However, recently rather regular multidecadal patterns of internal variability have been found in strongly eddying ocean models in the Southern Ocean, the so-called Southern Ocean Mode (SOM, Le Bars et al. (2016)). Dynamical processes involving eddy mean-flow interaction and eddy-topography interaction (Hogg and Blundell, 2006) are the driving mechanisms of the SOM (Jüling et al., 2018). Recently, van Westen and Dijkstra (2017) have shown that the SOM variability also occurs in a high-resolution ocean-eddying version of the CESM.

In this paper, we pursue the idea of the possible existence of a preferred multidecadal time scale of MRP events in the Southern Ocean through preconditioning and subsequent convection, by analysing the results of an extended CESM simulation as in van Westen and Dijkstra (2017). In this multi-century CESM simulation, several MRP events are found and we analyse the processes involved in these events. In Section 2, information on the CESM simulation and mean properties of the (Antarctic) sea-ice field in the simulation are given. As the Southern Ocean Mode is a relatively recent concept, we provide a brief overview and several characteristics of this mode in Section 3. Analysis of the MRP events in CESM is provided in Section 4. A summary and discussion of the results with the main conclusions are given in the final Section 5.

## 2   Climate Model and Methods

The model output of CESM version 1.0.4 (Hurrell et al., 2013) is taken from the same simulation as used in van Westen and Dijkstra (2017), which has been extended for this study to model year 250. The ocean component (POP, Parallel Ocean

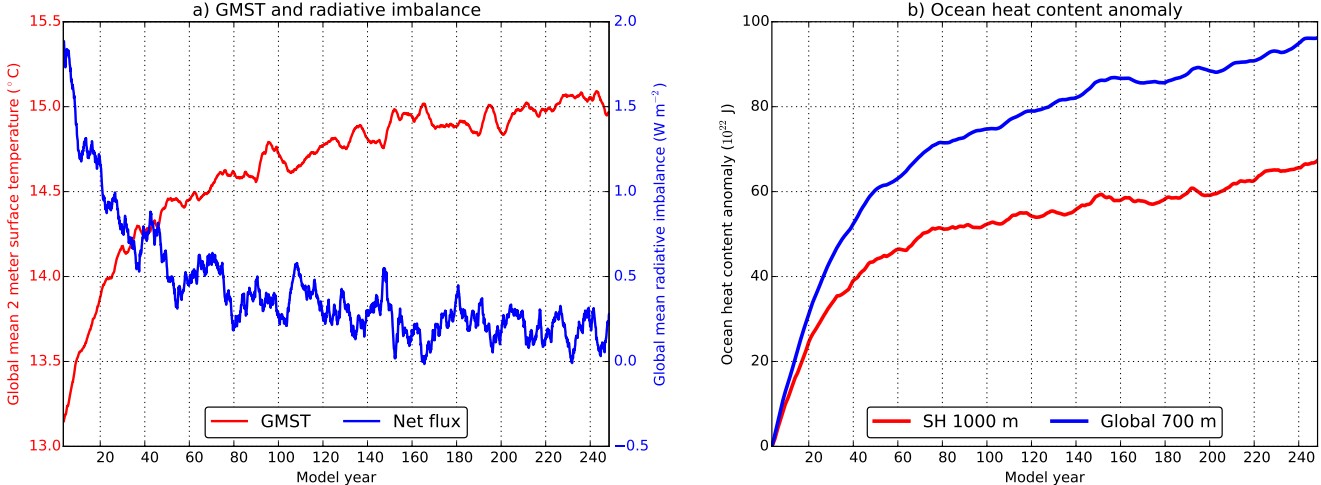

**Figure 1.** Equilibration of the CESM simulation for (a): the global mean 2 meter surface temperature (GMST) and global radiative imbalance at the top of the model (Net flux) and in (b): the upper 700 m global ocean heat content (OHC) anomaly and the upper 1000 m Southern Hemispheric OHC anomaly. The anomalies are with respect to their initial value of model year 1. All time series are smoothed by a 60-month running mean.

Program) and the sea-ice component (CICE) of the model have a $0.1°$ horizontal resolution on a curvilinear, tripolar grid which captures the development and interaction of mesoscale eddies (Hallberg, 2013). The ocean model has 42 non-equidistant depth levels, with the highest vertical resolution near the surface. The atmosphere- and land surface components of CESM have a horizontal resolution of $0.5°$, and the atmosphere component has 30 non-equidistant pressure levels. The forcing conditions
(e.g. $CO_2$, solar insulation, aerosols) are the observed ones over the year 2000 and are seasonally varying and repeated for every model year.

    The equilibration of the CESM simulation over the first 250 model years is shown in Figure 1 (these results extend those in van Westen and Dijkstra (2017)). Transient effects dominate the global mean (2 meter) surface temperature and global mean radiative imbalance at the top of the atmosphere time series during the first 100 model years (Figure 1a). These transient
effects become smaller after about 150 years as the CESM further adjusts to the present-day forcing conditions. The radiative imbalance remains slightly positive ($\sim 0.2$ W m$^{-2}$) over the last 100 years of the simulation. The upper 700 m global ocean heat content and upper 1000 m Southern Hemispheric ocean heat content are still adjusting (Figure 1b), but relatively small trends (compared to the first 100 model years of the simulation) occur over the last 100 years which can easily be removed through a quadratic detrending. The deep ocean fields take a much longer time to equilibrate and hence the model state is
not yet in equilibrium at year 250. Here, we analyse only the last 101 years (model years 150 – 250) of the simulation, using monthly averaged fields of the high-resolution output on the $0.1° \times 0.1°$ horizontal grid, instead of the interpolated output used in van Westen and Dijkstra (2017).

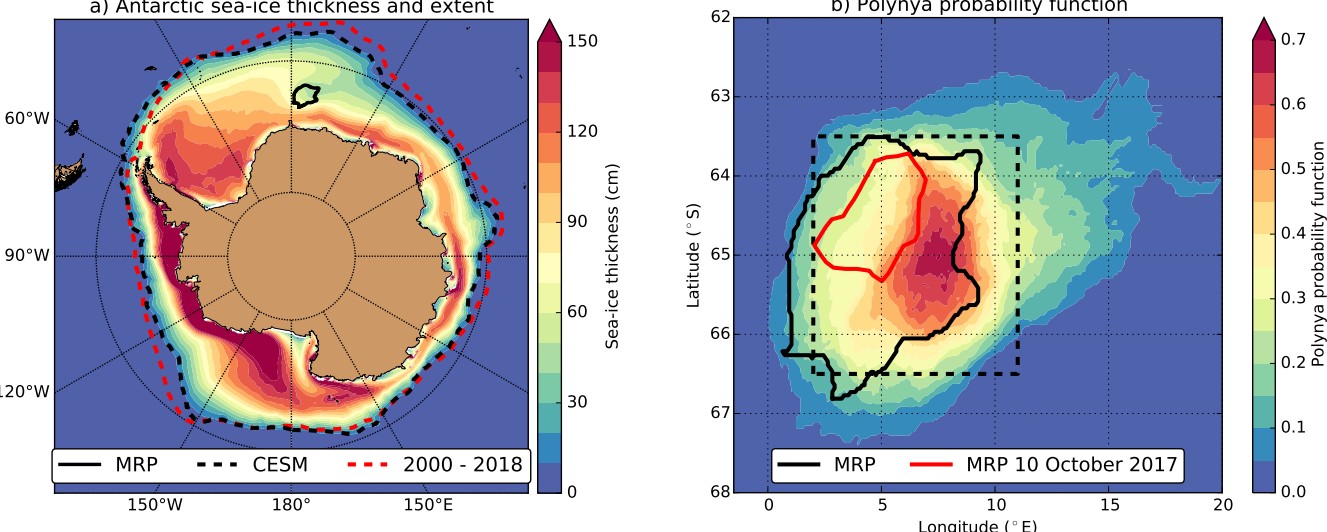

**Figure 2.** (a): Average sea-ice thickness (contours) for September over all the analysed years (model years 150 – 250) of the CESM output, including the time-mean September sea-ice extent (black dashed curve) and the time-mean (2000 – 2018) September sea-ice extent for SSMR-SSM/I. The September model year 181 MRP is shown by the black contour in a) and b). (b): Polynya probability function (explained in Section 2) based on all the polynya years. The dashed outlined region ($2°E - 11°E \times 66.5°S - 63.5°S$) is defined as the Polynya region. The observed 2017 MRP is indicated by the red contour in b).

The time-mean September sea-ice extent (based on a minimum sea-ice fraction of 15%) is shown in Figure 2a as the black dashed curve and the modelled sea-ice extent is slightly lower compared to observations (red dashed curve). The modelled and observed September sea-ice areas vary between 15.8 – 18.4 (101 model years) and 18.8 – 21.3 (19 years) million km$^2$, respectively. For observations we used sea-ice measurements by the Scanning Multichannel Microwave Radiometer (SMMR)

and Special Sensor Microwave Imager (SSM/I) (http://nsidc.org/data/G02202, Peng et al. (2013); Meier et al. (2017)). The time-mean September sea-ice thickness (colour plot in Figure 2a) displays relatively small values in the eastern part of the Weddell Sea. In this part of the Weddell Sea multiple polynyas over Maud Rise appear in the CESM; a modelled MRP is shown by the black curve in Figure 2a. In a companion CESM simulation, with a lower ocean resolution and sea-ice resolution of 1°, no MRP formation is found over a simulated period of 1300 years (results not shown).

The MRP is defined as the enclosed region within the Antarctic sea-ice pack that has a sea-ice fraction smaller than 15% (Weijer et al., 2017). The observed 2017 MRP and a CESM modelled MRP are both located over Maud Rise (Figure 2b). The MRPs found in CESM turn out to be quite localised in space and to determine that region, we computed a polynya probability density function. First, we define a polynya year to be a year where an MRP appears for at least three months. Secondly, at each MRP, the open-water grid cells (bounded by the 15% sea-ice fraction contour) and sea-ice covered grid cells are given a

label of 1 and 0, respectively. This procedure is applied to each polynya year, so non-polynya years are discarded. Taking the average over all the labeled fields (0 or 1), results in the polynya probability density function as shown in Figure 2b. There is

a small region near 7.5°E and 65°S where there is open water for almost all MRPs. Based on the polynya probability density function, we define the Polynya region (2°E – 11°E × 66.5°S – 63.5°S, shown as the dashed outlined region in Figure 2b).

We determined spatial averages over the Polynya region for the temperature, salinity, ocean heat content, sea-ice fraction, sea-ice thickness and the monthly maximum mixed layer depth. The mixed layer depth is defined as the depth at which the interpolated buoyancy gradient matches the maximum buoyancy gradient (this is standard output in CESM, Smith et al. (2010)). The local ocean heat content (OHC), indicated by $H_k$, at model level $z_k$ was calculated as:

$$H_k = \rho_k C_{p,k} T_k dz_k \tag{1}$$

where $dz_k$ is the vertical cell length of the model grid at that level. The quantities $\rho_k$, $T_k$ and $C_{p,k}$ are the local density, temperature and heat capacity, respectively. The temperature, salinity and pressure dependency are taken into account when calculating the local density and heat capacity (Millero et al., 1980; Sharqawy et al., 2010). The drift in the integrated OHC over the Polynya region (not shown) is smaller compared to that in the time series in Figure 1b.

The propagation of particles into the Polynya region is studied using OceanParcels (Probably A Really Computationally Efficient Lagrangian Simulator, version 2.1.4, Delandmeter and Van Sebille (2019)). In OceanParcels, one can release a set of virtual particles and track their path under the flow using Lagrangian modelling. Particles are passively advected by the time-varying 3D velocity field from the CESM output, either forward or backward in time. When a particle is released, its age is set to zero. We used a time step of $\Delta t = 1$ hour to update the location of each particle (by linear interpolation of the velocity fields, Delandmeter and Van Sebille (2019)) and the output of each particle (i.e. age, depth, latitude and longitude) is stored every 10 days.

## 3 The Southern Ocean Mode

In a stand-alone ocean simulation with the POP (not to confuse with the coupled model simulation, i.e. the CESM), Le Bars et al. (2016) identified a large-scale intrinsic mode of variability in the Southern Ocean (i.e. the SOM). In this simulation, the POP was forced by a yearly-repeated and seasonally varying atmosphere where the atmospheric forcing conditions were retained from climatology (see Le Bars et al. (2016) for more details). In Figure 3a we show the first Empirical Orthogonal Function (EOF) of the upper 1000 m averaged temperature field for the stand-alone POP (model years 175 – 275). Each time series of the temperature field was quadratically detrended, the seasonal cycle was removed and then it was normalised by its standard deviation. The resulting time series was then scaled by the area of the grid cell with respect to the largest grid cell area (which is set to 1) and then processed by principle component analysis (PCA; Preisendorfer (1988)). The corresponding Principal Component (PC) is shown in Figure 3c (red curve). The first EOF and PC capture 16% of the total variance and the dominant period of the SOM is 40 – 50 years (Le Bars et al., 2016; van Westen and Dijkstra, 2017; Jüling et al., 2018).

The variability associated with the SOM can also be measured via the SOM index which is defined as the surface temperature anomaly over the region 0°W – 50°W × 50°S – 35°S (Le Bars et al., 2016). The SOM index (also shown in Figure 3c) displays the same multidecadal variability as the first PC and has a phase difference of about 5 years (65 months) with respect to the

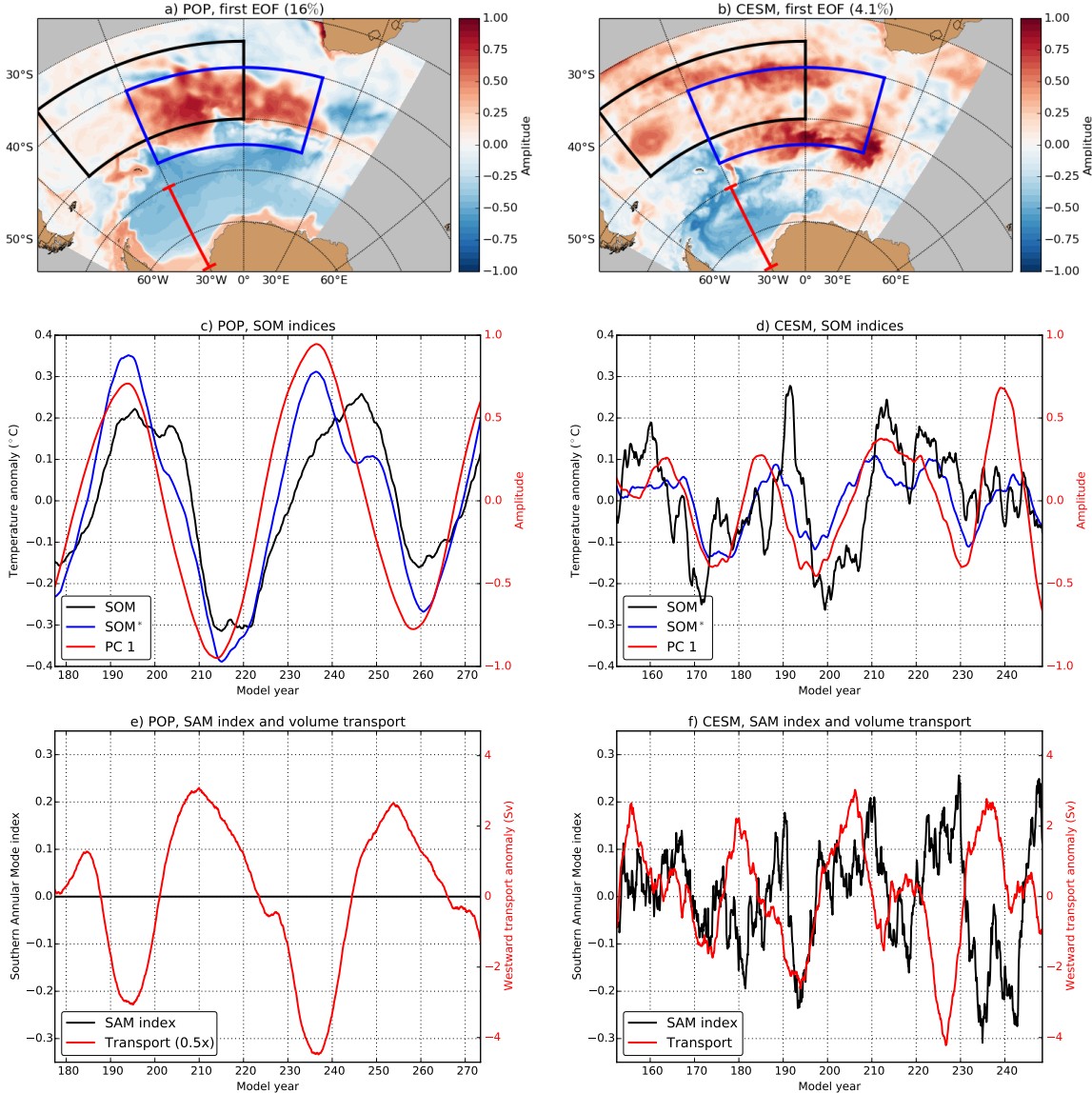

**Figure 3.** (a & b): The first EOF of the upper 1000 m averaged temperature in the Southern Ocean ($70°W – 40°E × 80°S – 30°S$) for the stand-alone POP and for the CESM. The black (blue) outlined region is used to determine the SOM (SOM*) index. The full-depth westward volume transport is determined over the red section ($30°W$, $77°S – 60°S$). (c & d): The SOM index, SOM* index and the first PC (of the upper 1000 m averaged temperature field) for the stand-alone POP and CESM. (e & f): The SAM index and westward volume transport anomaly at $30°W$ for the stand-alone POP and CESM. The time series in c) – f) are quadratically detrended and smoothed with a 60-month running mean. The anomalies of the westward transport are reduced by a factor 2 for the stand-alone POP.

first PC. The lag-correlation at 65 months is $r = 0.93$ and is significant (95%-confidence level (Cl from now on), taking into account the reduction of the degrees of freedom due to the running mean). The EOF dipole pattern affects the isopycnal slopes in the Southern Ocean (Le Bars et al., 2016) and consequently, via baroclinicity, the Weddell Gyre strength varies with the same period of the SOM (Figure 3e). The first PC and the Weddell Gyre strength time series are $180°$ out of phase and have a significant (95%-Cl) correlation of $r = -0.90$. The SOM is not related to the SAM (Gong and Wang, 1999) as the SAM index is constant in the stand-alone POP (Figure 3e).

Applying the same PCA above for the CESM output, we also find a large-scale pattern of variability in the Southern Ocean (Figure 3b). The dominant period of variability of the first PC is 25 years (Figure 3d, see also Figure 5b). The SOM index also displays multidecadal variability in the CESM (Figure 3d), but is much more variable compared to that of the stand-alone POP. For example, the lag-correlation between the first PC and the SOM index is much lower in the CESM ($r = 0.47$, 95%-Cl, 36 months) compared to the stand-alone POP ($r = 0.93$, 95%-Cl, 65 months). The differences in the EOF pattern and SOM index between the stand-alone POP and the CESM are related to atmospheric and sea-ice processes in the CESM. Therefore we propose the SOM$^*$ index, which is defined as the upper 1000 m averaged temperature anomaly over the region $30°W – 20°E \times 55°S – 40°S$ (blue outlined region in Figures 3a,b). The SOM$^*$ region is based on the maxima of the EOF patterns of both the stand-alone POP and CESM.

The multidecadal variability in the CESM is better represented in the SOM$^*$ index than in the SOM index (Figure 3d). For example, from lag-correlation analysis we find that the first PC is highly correlated with the SOM* index ($r = 0.80$, 95%-Cl, 19 months). For the stand-alone POP, the phase difference between the first PC and the SOM$^*$ index is 29 months and $r = 0.95$ (95%-Cl). Changing the the SOM$^*$ region (for example, to $0° – 20°E \times 50°S – 40°S$) did not significantly influence the results; it only increases the phase difference between the PC and the SOM* index. The Weddell Gyre strength also varies at the same 25-year period in the CESM (Figure 3f), where the first PC leads by 91 months ($r = -0.61$, 95%-Cl).

Atmospheric and sea-ice related processes also cause that less variance is captured in the first PC for the CESM (4.1%) compared to the stand-alone POP (16%). The first and second PC are well separated for both the stand-alone POP and CESM (North et al., 1982). Still, the variances are rather low in both models. Low variances in PCA are related to spatial noise in large parts of the ocean, the total number of grid cells and the length of the time series. Part of the spatial noise can be removed when the temperature fields are smoothed (through a 60-month running mean) before applying PCA. In this case, the captured variance by the first PC increases to 28.3% (stand-alone POP) and 15.3% (CESM). Another possibility to increase the variance is by reducing the region ($50°W – 30°E \times 70°S – 40°S$) over which PCA is applied; this results in a variance of 31.3% and 7.8% of the first PC for the stand-alone POP and CESM, respectively. Smoothing the time series in this smaller region further increases the variance to 47.1% (POP) and 23.1% (CESM). A similar large-scale pattern of multidecadal variability emerges in the first EOF (as in Figures 3a, b) when reducing the size of the region or by applying a moving average (not shown). To test whether the variances of PCA of the full field are significantly different from red noise (null hypothesis), we generated red-noise surrogate temperature fields with the same statistical properties as the pre-filtered time series. The variance of the first PC of these surrogate temperature fields is 3.8% and 0.9% at the 99%-Cl (500 surrogate fields) for the stand-alone POP

and CESM, respectively. Hence, although the variances of the PCs are relatively small in both the stand-alone POP and CESM, they are significantly different from red noise.

The SAM index (Gong and Wang, 1999) time series in the CESM (Figure 3f) displays inter-annual variability and does not match with that of the SOM related multidecadal variability. Lag-correlation analysis of the (smoothed) time series between the first PC and the SAM index did not result in any significant lag-correlations. Applying the lag-correlation analysis to the non-smoothed time series, non-significant and low values of lag-correlations ($|r| < 0.13$) are found between the first PC and the SAM index, as well as between the two SOM indices with the SAM index. Hence, in the CESM, atmospheric variability such as the SAM is not related to the large-scale pattern of multidecadal variability associated with the SOM.

Changes in the vertical stratification lead to a different SOM period. van Westen and Dijkstra (2017) find in the first 200 years of the CESM simulation a SOM period of about 40 – 50 years, which is a similar period to that found in the stand-alone POP of Le Bars et al. (2016). The meridional slope of the isopycnals increases (near $50°$S) over time in the CESM simulation (not shown), which enhances the baroclinic flow and reduces the period of the SOM. The background density profiles in the CESM output also have a larger meridional isopycnal slope compared to the stand-alone POP (not shown) and hence one would indeed expect a longer period in stand-alone POP compared to CESM. The zonal thermal-wind component near $50°$S in the CESM is about 1.6 times stronger compared to that of the stand-alone POP.

The SOM has not yet been reported in other modelling studies. One of the key features of the SOM is the mesoscale interaction with the background flow (Jüling et al., 2018). As most climate model simulations have a typical horizontal ocean resolution of about $1°$, mesoscale processes are not resolved (Hallberg, 2013) and therefore the SOM mechanism is absent in a non-eddying version of the POP (Le Bars et al., 2016). Moreover, long simulations ($> 100$ years) are required to develop the SOM in the models. In the first 50 years in the stand-alone POP (model years 75 – 125), the SOM related multidecadal variability is much weaker compared to remaining part of the simulation (van Westen and Dijkstra, 2017). For high-resolution models which resolve mesoscale processes (such as in Dufour et al. (2017) and Kurtakoti et al. (2018)), the simulated period is too short to develop and/or capture sufficient cycles of the SOM variability.

## 4 Results

In subsection 4.1, we first present the characteristics of the MRP events found in the CESM simulation. It turns out that there is a preferred multidecadal variability in MRP formation, possibly linked to the SOM (Le Bars et al., 2016), and this teleconnection is analysed in subsection 4.2. The processes causing the associated convective events, in particular the preconditioning of the density field are analysed in the last subsection 4.3.

### 4.1 Maud Rise Polynyas in the CESM

The September sea-ice fraction (blue curve in Figure 4a) shows relatively low values with respect to the time-mean for four periods: model years 158 – 159, 178 – 182, 205 – 209 and 231 – 237. These relatively low sea-ice fractions (as well as for the sea-ice thickness) are related to polynya formation over Maud Rise. The black curves in Figures 4b, c, d represent the annual

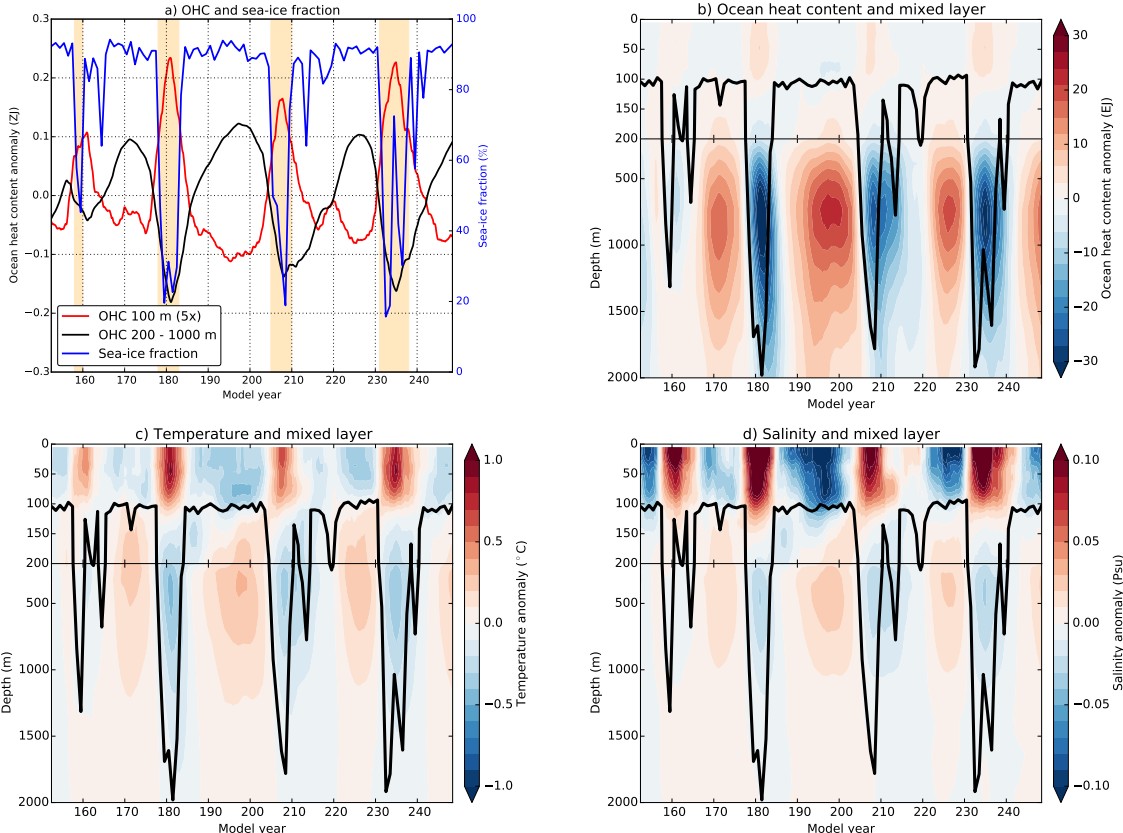

**Figure 4.** (a): Time series of the upper 100 m OHC anomaly ($\bar{H}_{100}$, magnified by a factor 5), the 200 – 1000 m OHC anomaly ($\bar{H}_{1000}$) and the September sea-ice fraction averaged over the Polynya region. The OHC time series are quadratically detrended and smoothed by a 60-month running mean. The shading indicates the polynya years. (b): Hovmöller diagram of the OHC anomaly vertical distribution (anomaly of OHC with respect to the time mean) over the Polynya region, smoothed by a 60-month running mean. The black curve indicates the annual maximum mixed layer depth. (c & d): Same as b), but now for the oceanic c) temperature and d) salinity.

maximum of the monthly maximum mixed layer depth averaged over the Polynya region. During MRP events, the mixed layer depth strongly increases over the Polynya region. A deepening of the mixed layer is also strongly correlated ($r = -0.98$) with a relatively low September sea-ice fraction over Maud Rise.

When the mixed layer deepens (i.e. during MRPs), heat and salinity anomalies are mixed towards the surface. The vertical
5   distributions for the OHC, temperature and salinity anomalies over the Polynya region are displayed in Figures 4b, 4c and 4d, respectively. To determine these anomalies at depth $z_k$, first the time mean (over model years 150 – 250) is subtracted from the time series and the result is quadratically detrended. The resulting time series ($H_k$, $T_k$ and $S_k$ for heat content, temperature and salinity, respectively) is smoothed using a 60-month moving average. Prior to each multiyear MRP event,

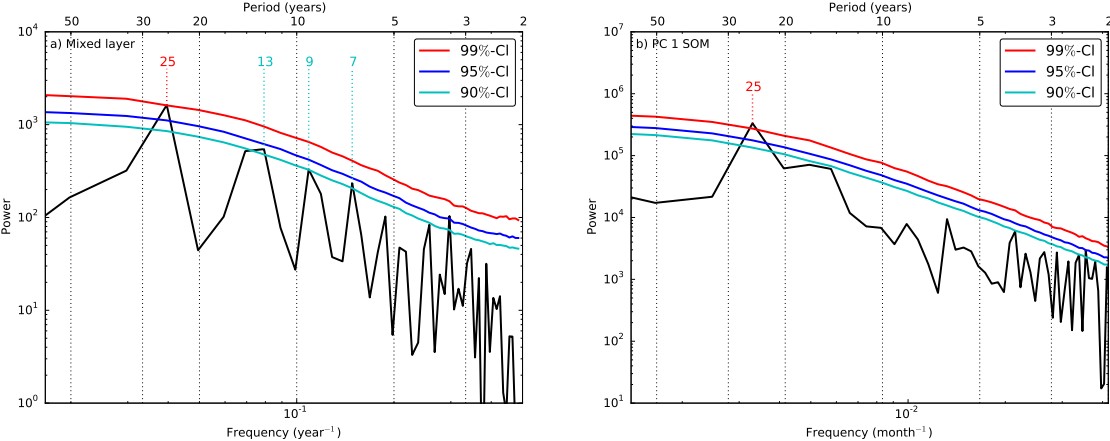

**Figure 5.** Fourier spectrum of the (a): mixed layer depth averaged over the Polynya region and (b): the SOM. Before determining the spectrum, the yearly mixed layer depth time series is normalised (to the standard deviation). For the SOM, we retained the non-smoothed first PC time series (see Section 3). The confidence levels (Cl) of significance (99% (red), 95% (blue) and 90% (cyan)) are derived from 10000 surrogate time series. The indicated periods (in years) are significant at the corresponding colour-coded confidence level, while only indicating low frequencies ($\geq 5$ years).

positive subsurface (200 – 2000 m) anomalies of OHC, temperature and salinity are found over the Polynya region (Figures 4b, 4c and 4d, respectively).

On the contrary, the surface layer (upper 100 m) remains relatively cold and fresh prior to polynya formation. This decoupling of the surface and subsurface is also shown in Figure 3a for $\bar{H}_{100}$ and $\bar{H}_{1000}$, where we integrated the OHC anomalies over the

upper 100 m and between 200 – 1000 m depths, respectively. The time series of $\bar{H}_{100}$ and $\bar{H}_{1000}$ are $180°$ out of phase. During an MRP event, the mixed layer depth increases, positive subsurface heat and salt anomalies are mixed towards the surface leading there to positive surface anomalies. The surface heat anomalies lead to sea-ice melt and consequently to polynya formation, in agreement with earlier studies (Martin et al., 2013; Dufour et al., 2017; Reintges et al., 2017).

Fourier spectra of the mixed layer depth and the first PC (cf. Figure 3d) are shown in Figure 5. The dominant period of

convection (i.e. deepening of the mixed layer depth) has a significant (99%-Cl) 25-year period against red noise (Figure 5a). The time series of OHC, temperature, salinity and sea-ice fraction vary with the same multidecadal period as a consequence of convection (spectra not shown). The first PC also varies on a dominant 25-year period which is significant (99%-Cl) against red noise (Figure 5b). The SOM and SOM* indices also display a dominant multidecadal period (20 – 30 years, not shown). The possible connection between the SOM and the mixed layer depth variability is analysed in the following section.

**4.2   SOM Related Variability in the Polynya Region**

As demonstrated in subsection 4.1, both the SOM and the mixed layer depth over the Polynya region display multidecadal variability with the same dominant time scale. In this subsection, the propagation of subsurface heat anomalies towards the

Polynya region is analysed. For this analysis, we use the SOM$^*$ index instead of the first PC because the first PC also contains variability due to MRP formation. The SOM$^*$ region is located far outside the Polynya region.

A lag-correlation analysis between the SOM$^*$ index (time series in Figure 3d), $\bar{H}_{100}$ and $\bar{H}_{1000}$ averaged over the Polynya region (time series in Figure 4a) is shown in Figure 6a. There are significant lag-correlations (95%-Cl) between the SOM$^*$ index

and $\bar{H}_{1000}$, where the SOM$^*$ index leads the $\bar{H}_{1000}$ by 7 years (80 months). At the similar time scale, negative correlations exists between $\bar{H}_{100}$ and the SOM$^*$ index. The first PC and the SOM index display similar significant lag-correlations with the $\bar{H}_{100}$ and $\bar{H}_{1000}$ averaged over the Polynya region.

The lag-correlation patterns of the SOM$^*$ index and the $\bar{H}_{1000}$ field (Figures 6b and 6c) at lag 0 and 24 months show overall positive and significant correlations in the South Atlantic, centered around the SOM$^*$ region. Positive correlations are

propagating along the southern part of the Weddell Gyre towards the Polynya region (Figures 5d – f). The first PC displays similar lag-correlations pattern with the $\bar{H}_{1000}$ field.

The time-mean flow near the Polynya region is westward and is set by the large-scale pattern of the Weddell Gyre (Figure 7a). Using OceanParcels (cf. section 2), one can determine the origin of the water mass of Maud Rise. Backtracking the particles (using the time-varying 3D velocity fields) shows that the particles mainly propagate along the Weddell Gyre and eventually

enter the Polynya region (Figure 7b). The particles are initially released at 200 m and 500 m depths in the Polynya region on December model year 250. One obtains different trajectories when the particles are released in, for example, December model year 249. Therefore, we released particles every 30 days in the Polynya region (each spaced by $0.5°$ and fixed initial depth, 133 particles per cast) and we backtracked these particles using OceanParcels over the CESM output (model years $150 - 250$).

The backtracked trajectories demonstrate that the particles propagate along the Weddell Gyre and have an upstream origin.

Rather than showing all the trajectories (as in Figure 7b), we show the particle distribution after 7 years (80 months) of backtracking (phase difference between the SOM$^*$ index and $\bar{H}_{1000}$ averaged over the Polynya region). The particle distributions are shown in Figure 8a, 8c and 8e where the particles are initially released at 200 m, 500 m and 1000 m depths in the Polynya region, respectively. These results indicate that most particles propagate (backwards) along the Weddell Gyre (similar as the trajectories in Figure 7b) with little dependence on the initial depth. Note that the final distributions overlap with the positive

correlations between the SOM$^*$ index and the $\bar{H}_{1000}$ field (Figures 6b,c). This implies that when the SOM$^*$ index (and also the SOM) is in a positive phase, positive subsurface OHC exists in these parts of the Weddell Gyre. These positive subsurface OHC anomalies propagate along the Weddell Gyre towards Maud Rise in about 7 years.

The backward propagation of the particles along the Weddell Gyre can also be identified using the colour-coded regions (as in Figures 8a,c,e). Here we determine the fraction of particles inside a specific region at a given age (Figures 8b,d,f). All

particles are released in the black-outlined region (this region slightly extends the Polynya region), and hence all particles have zero age. The time-mean flow in the black-outlined region is directed westwards (Figure 7a) which explains the relatively high ($\approx 70\%$) abundance of particles in the blue-outlined region after 2 years of backtracking. Via the blue dashed region, most ($> 50\%$) particles end up in the red region after about $9 - 13$ years.

The interpretation of these results is that when the SOM is in a positive phase, positive (subsurface) OHC anomalies are

found in the SOM$^*$ region (Figure 3b). The SOM$^*$ index, which is merely a measure of the phase of the SOM, shows significant

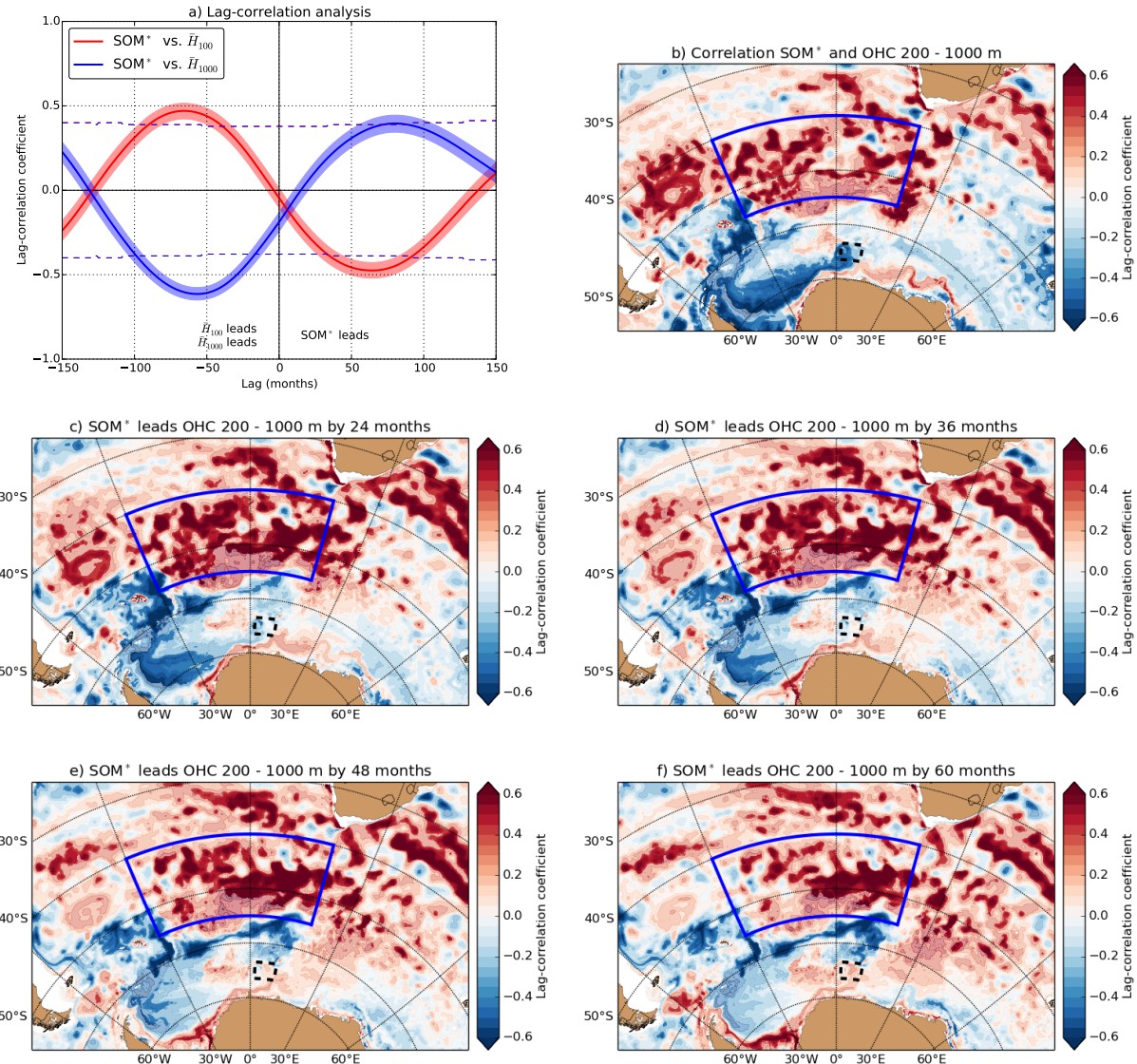

**Figure 6.** (a): Lag-correlation analysis of the SOM* index, the upper 100 m OHC anomaly ($\bar{H}_{100}$) and the 200 – 1000 m OHC anomaly ($\bar{H}_{1000}$), averaged over the Polynya region (see time series in Figure 3d and Figure 4a). A positive (negative) lag indicates that the SOM* time series leads (lags) the OHC time series. The shading indicates the 95%-confidence interval of lag-correlation, the dashed lines indicate the 95%-significant level. (b): Spatial lag-correlation pattern of the SOM* index time series and the $\bar{H}_{1000}$ field for zero lag. (c – f): Same as b), but now the SOM* index leads the $\bar{H}_{1000}$ field by c) 24, d) 36, e) 48 and f) 60 months. All time series are detrended and smoothed by a 60-month running mean before determining the lag-correlation. The blue outlined region is the SOM* index region and the dashed outlined region the Polynya region. The darker coloured correlations indicate significant (95%-CI) lag-correlations.

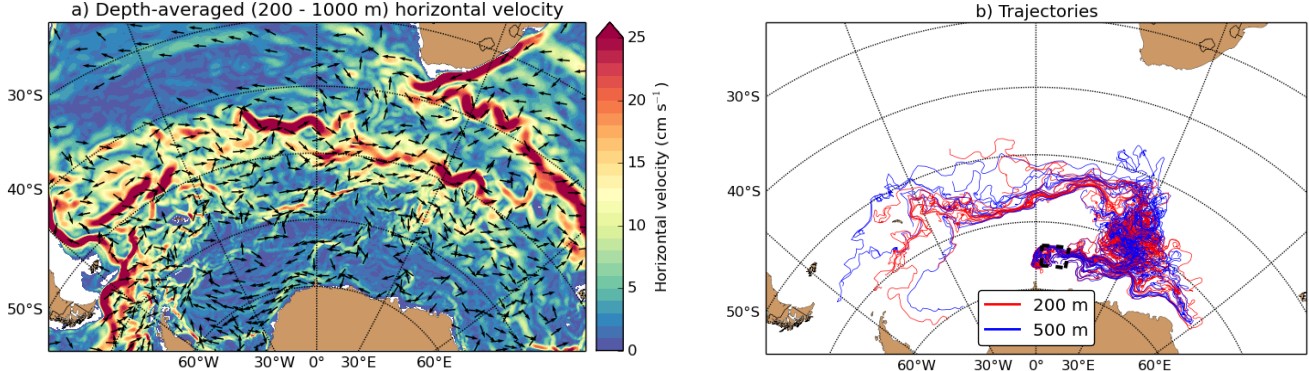

**Figure 7.** (a): Time-mean (model years 150 – 250) and depth-averaged (200 – 1000 m) horizontal velocity field (arrows), with the speed indicated in the contours. The arrows indicate the direction of the flow (not to scale) and are only shown for speeds larger than 2 cm s$^{-1}$. (b): The backtracked trajectories for 10 years for some particles (initially spaced at $1° \times 1°$, 30 particles per initial depth level) which are initially released at 200 m (red curves) and 500 m (blue curves) depths in the Polynya region (dashed outlined region).

correlations with the $\bar{H}_{1000}$ field and also positive correlations are found in the SOM$^*$ region (Figure 6b). These subsurface OHC anomalies reach the Polynya region in about 7 years (Figures 6d – f). In this way, the subsurface OHC anomalies are advected along the Weddell Gyre and enter the Polynya region (Figures 7 and 8).

### 4.3 Preconditioning of the Polynya region

The propagation of subsurface OHC anomalies associated with the SOM causes subsurface heat accumulation in the Polynya region and leads to changes in the subsurface heat reservoir. The horizontal advective heat flux, indicated by $\boldsymbol{F}_k$, at model level $z_k$, was calculated as:

$$\boldsymbol{F}_k = \rho_k C_{p,k} \boldsymbol{u}_k T_k \ , \tag{2}$$

where $\boldsymbol{u}_k$ is the horizontal velocity vector at depth $z_k$. The zonal heat input at the eastern boundary ($F_k^{\text{East}}$) of the Polynya
region (i.e. the normal component of $\boldsymbol{F}_k$ multiplied by the area perpendicular to the normal) with depth also displays multidecadal variability (Figure 9a). At mid-level depths, heat is advected into the Polynya region with the largest magnitudes occurring at depths between 700 – 900 m. This maximum in heat advection between 700 – 900 m depths is related to the maximum westward velocity which also occurs at subsurface depths. Positive (negative) values of $F_k^{\text{East}}$ indicate heat advection into (out of) the Polynya region.

The subsurface heat reservoir increases when there is net heat input into the Polynya region. The development of heat input between 200 – 1000 m depths over the four boundaries (i.e. $\bar{F}_{1000}^{\text{North}}$, $\bar{F}_{1000}^{\text{East}}$, $\bar{F}_{1000}^{\text{South}}$ and $\bar{F}_{1000}^{\text{West}}$) prior to the last multiyear MRP event is shown in Figure 9c. The net heat input between 200 – 1000 m depths (i.e. sum over the four boundaries, $\bar{F}_{1000}^{\text{Net}}$) is indicated by the black curve in Figure 9c and the net heat input over the upper 100 m is indicated by $\bar{F}_{100}^{\text{Net}}$ (black dashed curve). In addition, the temperature difference between subsurface ($\bar{T}_{1000}$, 200 – 1000 m depths) and surface ($\bar{T}_{100}$, upper

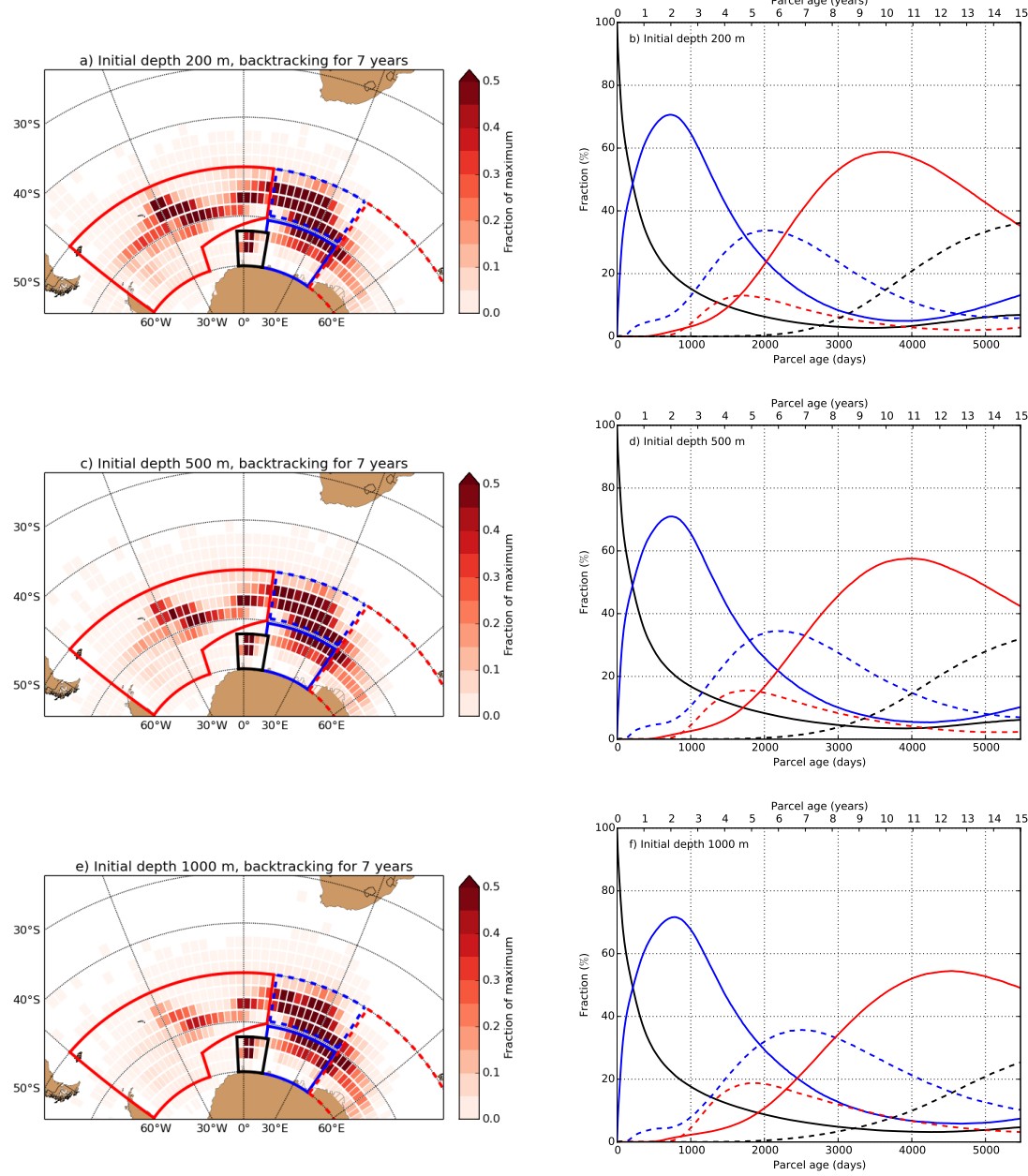

**Figure 8.** (a): Distribution of particles in the Southern Ocean after backtracking the particles for 7 years (80 months). The particles are initiated in the Polynya region at 200 m depth every 30 days. The colour-coded regions are used in b), where the red dashed region has a zonal extent of 60° (40°E – 100°E). (b): The time evolution of the fraction of particles in the colour-coded regions (see a)). The black dashed curve are particles outside the defined regions. (c – f): Similar as a) and b), but the particles are initially released at 500 m and 1000 m depths in the Polynya region.

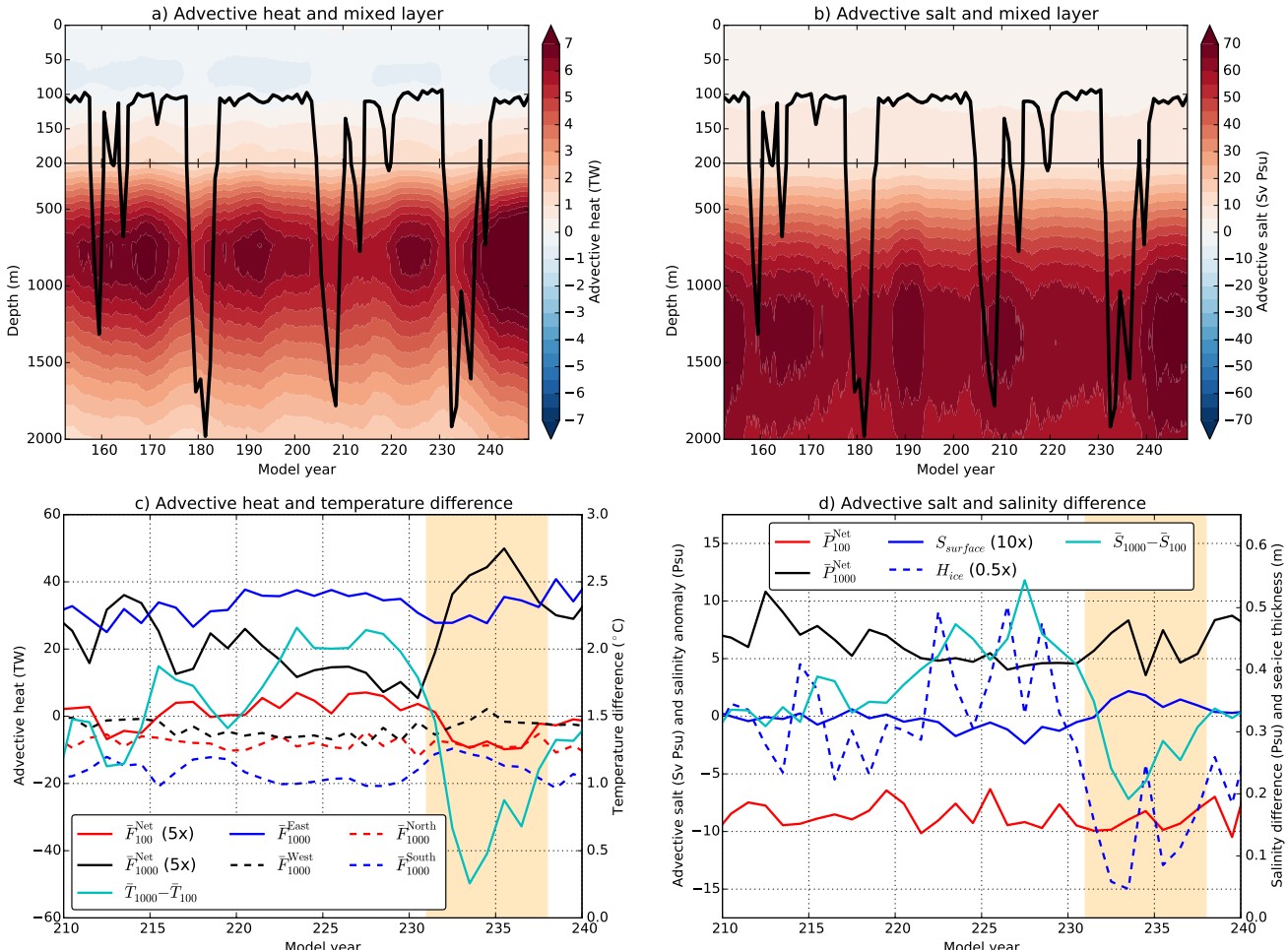

**Figure 9.** (a): Hovmöller diagram of the vertical distribution of heat input along the eastern boundary of the Polynya region ($11°$E, $63.5°$S – $66.5°$S). Positive (negative) values indicate heat advected into (out of) the Polynya region. The time series are smoothed by a 60-month running mean. The black curve again indicates the (annual maximum) mixed layer depth of the Polynya region. (b): Similar to a), but now for the salt input. (c): The horizontal heat input over the Polynya region boundaries over the model years $210 – 240$, where positive (negative) values of $\bar{F}$ indicate accumulation (depletion) of heat in the Polynya region over the corresponding vertical extent. The net horizontal heat input over the Polynya region boundaries are indicated by $\bar{F}^{\text{Net}}$ (magnified by a factor 5) over the corresponding vertical extent. The cyan curve is the temperature difference between subsurface ($\bar{T}_{1000}$) and surface ($\bar{T}_{100}$) over the Polynya region. All the time series consists of yearly averages and the shading indicates the last multiyear MRP event. (d): Similar to c), but now for the net advective salt input ($\bar{P}^{\text{Net}}_{100}$ and $\bar{P}^{\text{Net}}_{1000}$). The cyan curve is the salinity difference between subsurface ($\bar{S}_{1000}$) and surface ($\bar{S}_{100}$) over the Polynya region. In addition, the blue curve shows the yearly-averaged surface salinity anomaly with respect to model year 210 ($S_{surface}$, magnified by a factor 10) and the dashed blue curve displays the annual maximum sea-ice thickness ($H_{ice}$, reduced by a factor 2).

100 m) averaged over the Polynya region is shown in Figure 9c, which is a measure for the magnitude of the subsurface heat reservoir.

In model years 210 – 215, shortly after the third multiyear MRP event, the heat reservoir is depleted and the temperature difference is relatively small compared to period (model years 225 – 230) before the 231 – 237 MRP. Between model years 210 – 230, $\bar{F}_{1000}^{\mathrm{Net}}$ is positive and larger compared to $\bar{F}_{100}^{\mathrm{Net}}$, leading to an increase in the vertical temperature difference. As the temperature difference increases, the total amount of subsurface heat advected out (sum of $\bar{F}_{1000}^{\mathrm{North}}$, $\bar{F}_{1000}^{\mathrm{South}}$ and $\bar{F}_{1000}^{\mathrm{West}}$) of the Polynya region also increases, consequently the net heat input $\bar{F}_{1000}^{\mathrm{Net}}$ decreases over time. The time series (model years 150 – 250) of the temperature difference and $\bar{F}_{1000}^{\mathrm{Net}}$ are significantly anti-correlated (95%-Cl, $r = -0.88$). The build-up of the subsurface heat reservoir weakens the stratification over the Polynya region. During polynya formation, heat is vertically mixed causing a near zero temperature difference which indicates that the upper 1000 m is well mixed. Once convection ceases, the temperature difference returns to values of about $1.5°$C (which is comparable to the values in model year 210) and the build-up of the heat reservoir starts all over again. Observations prior to the MRP during the years 2016 – 2017 showed no long-term build-up of a subsurface heat reservoir (Campbell et al., 2019). However, there are observations near Maud Rise indicating subsurface heat accumulation over longer periods of time (Smedsrud, 2005; Fahrbach et al., 2004, 2011).

Preconditioning of the Polynya region also occurs by advection of salinity anomalies into the Polynya region and the advective salt fluxes are given by:

$$\boldsymbol{P}_k = \rho_k \boldsymbol{u}_k S_k . \tag{3}$$

The salt input at the eastern boundary ($P_k^{\mathrm{East}}$) of the Polynya region with depth is shown in Figure 9b. Here we expressed the salt input in units of Sv Psu (1 Sv $\equiv 10^6$ m$^3$ s$^{-1}$), where 1 Sv Psu is about $10^9$ g of salt per second. At the eastern boundary, salt is advected into the Polynya region over the upper 2000 m and the largest magnitude of salt advection is found between 1000 – 1500 m depths.

The eastern boundary contributes to subsurface salt accumulation in the Polynya region (Figure 9b), the other three boundaries (not shown) advect salt out of the Polynya region. The sign of $\bar{P}_{100}^{\mathrm{Net}}$ and $\bar{P}_{1000}^{\mathrm{Net}}$ are opposite, leading to an increase in the salinity difference between the subsurface ($\bar{S}_{1000}$, 200 – 1000 m) and surface ($\bar{S}_{100}$, upper 100 m) over the Polynya region prior to MRP formation (Figure 9d). An increase in the vertical salinity gradient, strengthens the stratification of the Polynya region. A saline surface layer (due to brine rejection) weakens the stratification, potentially leading to convection (Martinson et al., 1981). Therefore, we determined the surface salinity and sea-ice thickness averaged over the Polynya region which are also shown in Figure 9d. The yearly-averaged surface salinity time series is expressed as an anomaly with respect to model year 210. For the sea-ice thickness time series, we retained the annual maximum. Prior to MRP formation, the annual maximum sea-ice thickness is increasing over time leading to more brine rejection but this does not result in any increase of the surface salinity over the same period. Taking the average over the months April – July (i.e. sea-ice growth season) for the surface salinity also results in a decrease of surface salinity values prior to MRP formation. Changes in the vertical salinity gradient are mainly determined by horizontal (sub)surface advection of salt.

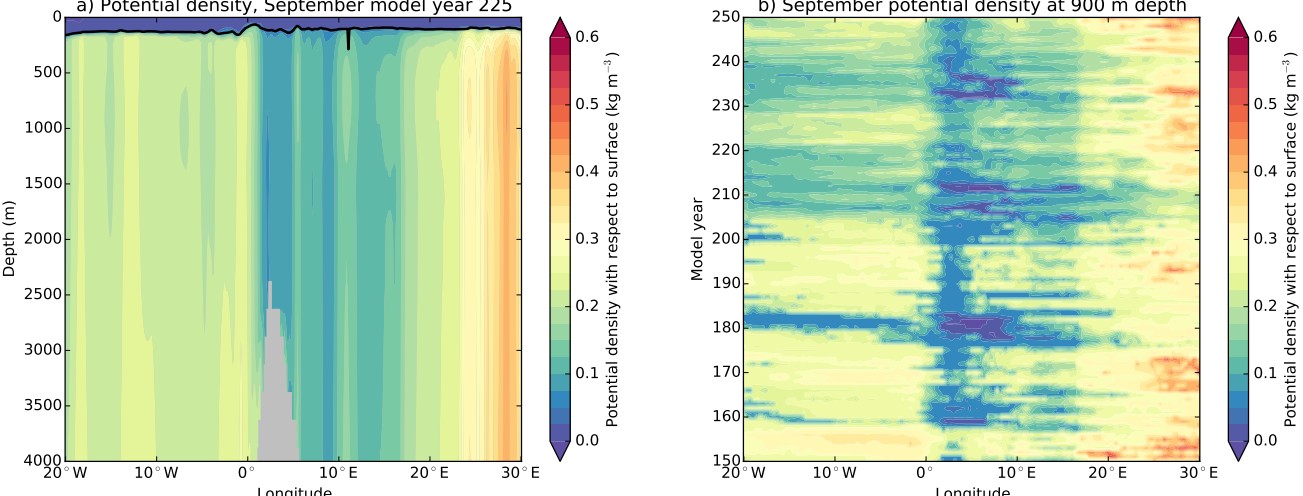

**Figure 10.** (a): Potential density across Maud Rise for September, model year 225. The black curve displays the monthly maximum mixed layer depth of that month. (b): Hovmöller diagram of the September potential density at 900 m depth. The potential density and maximum mixed layer depth are meridionally averaged between $65°S - 64°S$.

Stratified Taylor columns over Maud Rise also contribute to preconditioning of the region (Alverson and Owens, 1996; de Steur et al., 2007). The water column over Maud Rise is weakly stratified compared to the surroundings but there is no (deep) convection prior to MRP formation (Figure 10a). Here we determined the potential density (standard output of the CESM) and mixed layer depth between $65°S - 64°S$ and across $20°W - 30°E$; this section crosses the Maud Rise summit (near $3°E$). There
are also Taylor columns over Maud Rise during other years, as shown by the potential density at 900 m depth in Figure 10b. During the polynya years, the potential density at 900 m depth (as well as for other depth levels) tend to zero over Maud Rise indicating that the water column overturns. Gordon et al. (2007) suggest that a negative phase of the SAM contributes to the preconditioning of the region, however, the SAM is (strongly) positive before the first (158 – 159), third (205 – 209) and fourth (231 – 237) multiyear MRP event in the CESM (Figure 3f).
Based on this analysis of the CESM results, the preconditioning of the density field seems to be most affected by the subsurface heat anomalies and Taylor columns over Maud Rise. Positive subsurface heat and salt anomalies, advected via the Weddell Gyre, increase the vertical gradient in temperature and salinity over the Polynya region, respectively. The increased vertical temperature (salinity) gradient weakens (strengthens) the stratification near Maud Rise.

## 5    Summary and Discussion

We analysed the last 100 years of model output from a 250-year control simulation of a high-resolution version of CESM under a repeated seasonal forcing of the year 2000. We find four multiyear MRP events and, in each event, (deep) convection causes vertical mixing of anomalous subsurface heat towards the surface where it melts the sea ice and leads to the formation

of the MRP. These processes of the formation and lifecycle of the MRP are broadly in agreement with the classical view as in Martinson et al. (1981) and those described from other model results in Martin et al. (2013), Dufour et al. (2017) and Reintges et al. (2017). In most model studies deep convection occurs rather randomly while in the CESM deep convection occurs about every 25 years.

In the CESM, we find that the multidecadal preconditioning of the subsurface density can be linked to an intrinsic dynamical ocean mode in the Southern Ocean, the SOM, which is dominantly caused by eddy-mean flow interaction (Jüling et al., 2018). A positive phase of the SOM leads to positive subsurface OHC anomalies in the South Atlantic Ocean (Le Bars et al., 2016). These anomalies enter the Weddell Gyre near 30°E and eventually reach the Polynya region, where the anomalies cause deep convection and sea-ice melt. Hence, the frequency of occurrence of the MRP events is related to the SOM variability through
the advection of the subsurface OHC anomalies in the Weddell Sea. In this case, a preferred frequency of convective events is induced through preconditioning, which is around 25 years in the CESM.

Although preconditioning of the density field in the Maud Rise region is connected to subsurface advection of heat and salt anomalies and Taylor columns, it does not strictly imply polynya formation. However, it will favour the occurrence of MRPs due to atmospheric variability, such as intense winter storms. For example, the formation of the 2016 – 2017 MRP was initiated
by an increased storm frequency under the influence of a positive SAM index (Campbell et al., 2019). In the CESM, the SAM index is in a positive phase prior to MRP formation, but the initiation process of the MRPs is out of the scope of this study.

Observations are unfortunately too sparse to falsify the hypothesis of a preferred multidecadal variability in the Southern Ocean. Reanalysis of sea surface temperatures are available to derive a historical SOM index, but the time series is too short to adequately detect the SOM. Relatively low sea-ice fractions were observed over Maud Rise in the mid-1970s, 1980, 1994
and 2016 – 2017 (Comiso and Gordon, 1987; Lindsay et al., 2004; Gordon et al., 2007; Campbell et al., 2019), suggesting a 15 – 20 year return period of MRP formation which is a bit shorter compared to the 25-year period of the CESM MRPs. The background state of the Southern Ocean sets the re-occurrence time for deep convection (Reintges et al., 2017) which explains the differences in the return period of MRP formation. Besides, the stand-alone POP has a different background state compared to that of the CESM, leading to much longer re-occurrence time of deep convection in the stand-alone POP compared to the
CESM (Jüling et al., 2018). Nonetheless, if the view is correct that a preferred multidecadal frequency (~20 years) in the Southern Ocean exists, it would imply that a next MRP event, if not affected by climate change (De Lavergne et al., 2014), can be expected before the mid-2030s.

*Acknowledgements.* The authors thank Michael Kliphuis (IMAU, UU) for performing the CESM simulations. The computations were per-formed on the Cartesius at SURFsara in Amsterdam, the Netherlands. Use of the Cartesius computing facilities was sponsored by the
Netherlands Organization for Scientific Research (NWO) under the project 15552. The data from the model simulation used in this work is available upon request from the authors. The NOAA/NSIDC provided the satellite sea-ice products (http://nsidc.org/data/G02202). The OceanParcels framework can be obtained online (oceanparcels.org) and we thank Peter Nooteboom (IMAU, UU) for his assistance in setting up this code for the CESM data.

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
