# Peer review of "Multidecadal Preconditioning of the Maud Rise Polynya Region"

_Ocean Science, 2020_

## Referee Comment (RC1) · Anonymous Referee #1 · 22 May 2020

This paper explores the mechanism of the Maud Rise polynya in the CESM model. The MRP in CESM re-occurs quite regularly every ∼25 years. The authors argue that the MRP is linked to the Southern Ocean Mode, which provides the 25 years timescales. The connection between the 2 phenomena is achieved through advection of subsurface heat content anomalies from the southern Atlantic to the Maud rise region.

The study is overall interesting and well conducted. The obvious limitation is that this is a one model study, in a domain where models have shown little consistency. Nonetheless, it is worth of publication to provide a possible avenue to investigate other models or the real world.

The manuscript could be significantly strengthen by clarifying the link with the SOM and the mechanism setting the timescale.

[Figure]

Major comments:

1) Link with the Southern Ocean mode (SOM): - Please explain what is the SOM. It's not the NAO or the SAM. As far I can see it has been investigated in handful of papers, so most readers will have no idea what this is. - The connection between the MRP and the SOM is unclear. What the figures suggest (Fig.6b, Fig. 7) is a connection between the MRP and the area in the red box (50S-60S, centered on 30W). Then, the OHC in the red box is correlated with the SOM. It seems like a 2 step connection. Perhaps this reflects me not understanding what is the SOM, but what does the link with the SOM provides here?

2) The computation and interpretation of Pbrine are unclear. I understand why Pbrine is in Sv psu for comparison with the other fluxes, but could it be also given in m of sea ice (thickness). This would be a more useful measure.

Pbrine contributes to increase the salinity of the upper layer (during non-polynia years). From Fig. 3, salinity in the upper layer decreases during non-polynya years. So what is the point of developing so much the Pbrine diagnostics when it cannot explain the behavior observed in Fig. 3 and cannot be a major contributor to the balance? More generally, I do not get what is the main outcome of the salinity analysis. The salinity stratification delays the destratification over Maud Rise by working against the temperature changes?

3) Page 14: "Hence, the frequency of occurrence of the MRP events is related to the SOM variability through the advection of the subsurface OHC anomalies in the Weddell Sea. In this case, a preferred frequency of convective events is induced through preconditioning, which is around 25 years in the CESM."

This is probably the main point of the paper, but this needs clarification. Are you suggesting that the 25 years periodicity of the SOM set the 25year period of the MRP (with a 10-year advective delay)?

If so, why is it just the 25 year period that is selected as the SOM exhibits other peaks (at 34, 17, 5 yr in Fig. 4b, the 25 years peak is actually not outstanding). This suggests a selection mechanism. I could speculate that there must be a minimum threshold for the subsurface heat content build up in the MRP to trigger a convective event. So possibly 5 and 17 years are not long enough. If 25 years is enough to get to instability, the 34 years peak in SOM would never show up in the MRP. Let's assume 2C is the critical temp difference to get instability (Fig. 8c). Could we say that, at the rate of convergence seen in Fig. 8c (about20-10 TW) , it takes ∼25 years to build on a 2C difference?

My main point is that the argument cannot just be "a 25 year timescale in SOM translates into a 25 year timescale in MPR". There is another effect for selecting the timescale.

Going further, this selection mechanism may fully determine the timescale. That is, the SOM acts as a white noise providing variability on all timescale, no need for a peak in SOM. The build up of the subsurface OHC to instability requires 25 years, and only this frequency shows up in MRP.

Please clarify.

4) page 5, line 10: I do not understand why particles older than 10 years are removed. Please explain this choice. This is even more surprising that 10 years misses the most interesting peak seen for the red box in Fig. 7.

Minor points:

- Fig. 2: what is the thick black line in panel b)? Same as in panel a)?

- page 5, line 5: "The temperature, salinity and pressure dependency are taken into account when calculating the local density and heat capacity (Millero et al., 1980; Sharqawy et al., 2010)."

Does it make sense to do this? Does the model account for variations of density and

heat capacity? I don't know the details of CESM, but many models assume that the heat capacity is constant and also assume that, in the heat budget, the density is constant. Such that the heat content of a grid cell is rho_o Cp dz(k) T(k) (dz(k) might vary due to stretching of the vertical coordinate). So are you introducing an inconsistent and unnecessary complication?

- Fig. 6b: how long are the trajectories shown here? And how many of them are plotted?

- Fig. 3: Panel b is quite cluttered and difficult to decipher. Possibly you could remove the dashed black line. The information about the anti correlation between the upper- and lower-layer temperatures is quite obvious in panel c)

- Still Fig. 3: there is a shift between the panel a) and panel c). At first look, I thought there was a shit between the variables with lead-lag effects. Until I realize that panel c) is shifted relatively to panel a) because of the color bar. It would be convenient to line up the 2 panels.

- page 13, line 1: " is A measure"

―――――――――――――――――――――――

---

## Referee Comment (RC2) · Anonymous Referee #2 · 4 Jun 2020

Manuscript to review: Multidecadal Preconditioning of the Maud Rise Polynya Region

Authors: René M. van Westen and Henk A. Dijkstra

Reviewer decision: Accepted with Major Revisions.

The manuscript presents a study of the preconditioning mechanisms of Maud Rise Polynyas (MRPs) in a high-resolution CESM simulation. The four MRP events occur roughly 20-25 years apart. The polynyas' periodicity is attributed to the Southern Ocean Mode (SOM), a mode of ocean heat content (OHC) variability that is not validated with observations or theory. The study attributes the polynyas to be singularly preconditioned by the SOM, where they suggest that a positive phase of the SOM then leads to positive subsurface OHC anomalies in the South Atlantic ocean, which then

enters the Weddell gyre over ten years and causes deep convection over Maud Rise. The study is interesting, but the fundamental hypothesis is not backed up with a plausible physical explanation, nor do the authors justify why SOM is a unique mode to consider.

Major comments

1. What is the Southern Ocean Mode? There are very few modeling studies that even consider it. Do the authors consider it to be a different mode from the Southern Annular Mode (SAM)? However, one of the definitions of SAM index is the difference between normalized zonally mean sea level pressure between 40S and 65S (Gong and Wang 1999). (SAMindex = P*40S – P*65S ) whose Atlantic sector encompasses the region used in the definition of SOM. The authors do not provide an explanation as to why the SOM index is an independent mode. The Southern Annular Mode could in fact influence the Temperature anomaly over 0-50W;35-50S. It would be very helpful if the authors can show that the SOM index is different from the SAM index. It is really not clear why this is a better metric or how this region behaves as an independent influence on the preconditioning of the polynyas over Maud Rise.

2. The authors do not present a robust justification for using only the SOM and not looking at any of the important factors that play a role in preconditioning such as the structure and strength of the Weddell gyre (Cheon et al. 2015) , the SAM index (Gordon et al. 2007), or the presence of stratified Taylor columns over Maud Rise (de Steur et al. 2007; Alverson and Owens 1996). Previous literature (Hirabara et al. 2012) suggests that the preconditioning and formation mechanism of these MRPs are related to several contributing factors that involve the interaction of large- and small-scale ocean/atmospheric circulation. The authors here do not provide a clear physical connection on how the SOM can in turn influence these other contributing factors for polynya formation. The lack of a physical mechanism clearly linking the SOM to the MRPs is a major weakness in this publication.

3. The loosely connected interpretation of figures 5,6 and 7 suggests that particles released over the MRP box and backtracked for 8 years end up in specific regions in the Weddell gyre that are correlated to parts of the SOM region. However, correlation does not imply causation.

4. If the subsurface OHC anomalies from the SOM region propagate along the Weddell gyre to the MRP box in ten years, one should be able to see a pattern in the correlation between SOM and H1000 in which the SOM leads H1000 in steps of 12-24 months (similar to figure 5d but in steps where SOM leads H1000 in 24, 48, 72, 96 months). It would help readers to see the intermediate steps of SOM leading H1000 by 12-24 months between figures 5c and 5d in which the propagation should be evident if such connections exist.

Minor comments

1. The mapping projection in figure 5 is not consistent with figures 6-7. Please be consistent with the mapping projections.

2. The authors mention 3 MRP events observed in 1973-1977,1994 and 2016-2017 to suggest a periodicity of 20 years between MRP events. They did not include the MRP event of 1980 (Comiso and Gordon 1987). The observations do not suggest a periodicity of 20 years. The polynyas of the late-70s were extremely large Weddell Sea Polynyas that begin over Maud Rise and spread westward into the Weddell Sea and have not been observed since.

References

1. Alverson, K., and B. Owens, 1996: Topographic pre-conditioning of open-ocean deep convection. J. Phys. Oceanogr., 26, 2196–2213, doi:10.1175/1520-0485(1996)026<2196:TPOOOD>2.0.CO;2.

2. Cheon, W. G., S. Lee, A. L. Gordon, Y. Liu, C. Cho, and J. J. Park, 2015: Replicating the 1970s' Weddell Polynya using a coupled ocean-sea ice model with reanalysis

surface flux fields. Geophys. Res. Lett., 42, 5411–5418, doi:10.1002/2015GL064364.

3. Comiso, J. C., and A. L. Gordon, 1987: Recurring polynyas over the Cosmonaut Sea and the Maud Rise. J. Geophys. Res. Ocean., 92, 2819–2833, doi:10.1029/JC092iC03p02819.

4. Gong, D., and S. Wang, 1999: Definition of Antarctic oscillation index. Geophys. Res. Lett., 26, 459–462, doi:10.1029/1999GL900003.

5. Gordon, A. L., M. Visbeck, and J. C. Comiso, 2007: A possible link between the Weddell Polynya and the southern annular mode. J. Clim., 20, 2558–2571, doi:10.1175/JCLI4046.1.

6. Hirabara, M., H. Tsujino, H. Nakano, and G. Yamanaka, 2012: Formation mechanism of the Weddell Sea Polynya and the impact on the global abyssal ocean. J. Oceanogr., 68, 771–796, doi:10.1007/s10872-012-0139-3.

7. de Steur, L., D. M. Holland, R. D. Muench, and M. G. Mcphee, 2007: The warm-water "Halo" around Maud Rise: Properties, dynamics and Impact. Deep. Res. Part I Oceanogr. Res. Pap., 54, 871–896, doi:10.1016/j.dsr.2007.03.009.

---

## Author Comment (AC1) · 6 Jul 2020

**MS-No.:** os-2020-25

**Version:** Revision

**Title:** Multidecadal Preconditioning of the Maud Rise Polynya Region

**Author(s):** René M. van Westen and Henk A. Dijkstra

**Point-by-point reply to reviewer #1**

July 6, 2020

We thank the reviewer for his/her careful reading and for the useful comments on the manuscript.

Major comments:

1. *Link with the Southern Ocean mode (SOM): - Please explain what is the SOM. It's not the NAO or the SAM. As far I can see it has been investigated in handful of papers, so most readers will have no idea what this is. - The connection between the MRP and the SOM is unclear. What the figures suggest (Fig.6b, Fig. 7) is a connection between the MRP and the area in the red box (50S-60S, centered on 30W). Then, the OHC in the red box is correlated with the SOM. It seems like a 2 step connection. Perhaps this reflects me not understanding what is the SOM, but what does the link with the SOM provides here?*

   **Author's reply:**
   To provide more information on the relation between the SOM and SAM, we have already performed additional analysis. First, we analysed the SOM using the stand-alone POP simulation which was first used to detect the SOM (Le Bars et al. 2016) using a Principal Component Analysis (PCA) of the upper ocean

temperature fields in the Southern Ocean. A large-scale pattern of variability is found then with a period of about 40 – 50 years. In the stand-alone POP model, the associated PC displays the same multidecadal variability as the SOM index in Le Bars et al. (2016). This multidecadal variability is not related to any atmospheric variability since the atmospheric forcing is seasonally varying in the stand-alone POP.

Secondly, the same PCA (as above) is done for the CESM (fully coupled climate model). The first principal component displays a 25-year period and this period is significant against red noise in the CESM. The SOM index time series contains much more noise compared to the first principal component. The noise in the SOM index is likely related to atmospheric variability (which is absent in the stand-alone POP). Therefore, we do not use the SOM index in the revision anymore but the first principal component time series of the PCA.

Finally, we determined the SAM index in the stand-alone POP and in the CESM. The SAM index is constant in the stand-alone POP. For the CESM, the SAM index displays inter-annual variability over the 101-year period. In both the stand-alone POP and CESM results, we find no relation between the SOM and SAM.

**Changes in manuscript:**
We will include a new section in the revision: The Southern Ocean Mode. In this section, the PCA analysis is described and we motivate why the SOM is different from the SAM. In the revision, we will use the first principal component of the PCA to measure the SOM instead of the SOM index. All figures and results will be changed accordingly.

2. *The computation and interpretation of Pbrine are unclear. I understand why Pbrine is in Sv psu for comparison with the other fluxes, but could it be also given in m of sea-ice (thickness). This would be a more useful measure.*

*Pbrine contributes to increase the salinity of the upper layer (during non-polynia years). From Fig. 3, salinity in the upper layer decreases during non-polynya years. So what is the point of developing so much the Pbrine diagnostics when it cannot explain the behavior observed in Fig. 3 and cannot be a major contributor to the balance? More generally, I do not get what is the main outcome of the salinity analysis. The salinity stratification delays the destratification over Maud Rise by working against the temperature changes?*

**Author's reply:**

In the simple 1D-box model in Martinson et al. (1981), the water column overturns due to brine rejection. The salt budget analysis in CESM shows that brine rejection is not a major contributor to the balance and, from this, it is not likely that brine rejection leads to the overturning of the water column in the CESM.

**Changes in manuscript:**

We will rewrite and clarify this section in the revision. The Pbrine time series will be replaced by the sea-ice thickness time series. In addition, we will determine the surface salinity averaged over the Polynya region and include this in the revision. The results and figures will be changed accordingly.

3. *Page 14: "Hence, the frequency of occurrence of the MRP events is related to the SOM variability through the advection of the subsurface OHC anomalies in the Weddell Sea. In this case, a preferred frequency of convective events is induced through preconditioning, which is around 25 years in the CESM."*

*This is probably the main point of the paper, but this needs clarification. Are you suggesting that the 25 years periodicity of the SOM set the 25 year period of the MRP (with a 10-year advective delay)?*

*If so, why is it just the 25 year period that is selected as the SOM exhibits*

*other peaks (at 34, 17, 5 yr in Fig. 4b, the 25 years peak is actually not outstanding). This suggests a selection mechanism. I could speculate that there must be a minimum threshold for the subsurface heat content build up in the MRP to trigger a convective event. So possibly 5 and 17 years are not long enough. If 25 years is enough to get to instability, the 34 years peak in SOM would never show up in the MRP. Let's assume 2C is the critical temp difference to get instability (Fig. 8c). Could we say that, at the rate of convergence seen in Fig. 8c (about 20-10 TW) , it takes ~25 years to build on a 2C difference?*

*My main point is that the argument cannot just be "a 25 year timescale in SOM translates into a 25 year timescale in MPR". There is another effect for selecting the timescale.*

*Going further, this selection mechanism may fully determine the timescale. That is, the SOM acts as a white noise providing variability on all timescale, no need for a peak in SOM. The build up of the subsurface OHC to instability requires 25 years, and only this frequency shows up in MRP.*

*Please clarify*

**Author's reply:**
We agree that the 25-year period in the SOM index is not the dominant variability and that variability associated with the 'other peaks' in the SOM index could in principle also induce MRP formation (with a time delay) if the critical temperature difference is reached, as suggested by the reviewer. We have already conducted a PCA regarding the SOM (see also Major comment 1) in the CESM and we find that several of these 'other peaks' in the SOM index are likely related to atmospheric variability. The PC1 time series of the SOM clearly shows the dominant 25-year period which is significant against red noise (99%-CL).

**Changes in manuscript:**
We will include results of the PCA analysis of the SOM and use this PC1 instead of the SOM index (see also Major comment 1). We will update the results and figures accordingly. In addition, we will include more lag-correlation patterns between the SOM and the subsurface ocean heat content fields to demonstrate the propagation of heat anomalies towards Maud Rise.

4. *page 5, line 10: I do not understand why particles older than 10 years are removed. Please explain this choice. This is even more surprising that 10 years misses the most interesting peak seen for the red box in Fig. 7.*

**Author's reply:**
This is indeed a subjective choice. We have already backtracked the particles for a longer period of time. As suggested by the reviewer, we will extend the plots (to 15 years); the peak (9 – 12 years) for the red box is now visible for all three plots.

**Changes in manuscript:**
We will replace the earlier results by those in which the particles are backtracked for a longer period of time.

Minor comments:

1. *Fig. 2: what is the thick black line in panel b)? Same as in panel a)?*

**Author's reply:**
This is the boundary of the modelled MRP during September model year 181.

**Changes in manuscript:**
We will clarify this in the main text and in the caption of the figure.

2. *Page 5, line 5: "The temperature, salinity and pressure dependency are taken into account when calculating the local density and heat capacity (Millero et al.,*

*1980; Sharqawy et al., 2010)."*

*Does it make sense to do this? Does the model account for variations of density and heat capacity? I don't know the details of CESM, but many models assume that the heat capacity is constant and also assume that, in the heat budget, the density is constant. Such that the heat content of a grid cell is $\rho_0$ Cp dz(k) T(k) (dz(k) might vary due to stretching of the vertical coordinate). So are you introducing an inconsistent and unnecessary complication?*

**Author's reply:**
Due to storage limitations, the density was not written out in the first part of the simulation. Therefore, we have chosen to compute the density (using T,S,P) as mentioned in the manuscript. We have later verified these values of the density with the standard CESM density output and these values were similar. The density and heat capacity in the CESM are not constant but local variations in density and heat capacity are very small.

**Changes in manuscript:**
No changes in text.

3. *Fig. 6b: how long are the trajectories shown here? And how many of them are plotted?*

**Author's reply:**
The particles were backtracked for 10 years and the particles were initially released at a $1° \times 1°$ grid, which means a total of 30 particles per initial depth level.

**Changes in manuscript:**
We have revised the caption accordingly.

4. *Fig. 3: Panel b is quite cluttered and difficult to decipher. Possibly you could remove the dashed black line. The information about the anti correlation between the upper and lower-layer temperatures is quite obvious in panel c)*

**Author's reply:**
Suggestion followed.

**Changes in manuscript:**
The SOM index is shown in the new Section 'The Southern Ocean Mode' and is removed from Fig. 3.

5. *Still Fig. 3: there is a shift between the panel a) and panel c). At first look, I thought there was a shit between the variables with lead-lag effects. Until I realize that panel c) is shifted relatively to panel a) because of the color bar. It would be convenient to line up the 2 panels.*

    **Author's reply:**
    Suggestion followed.

    **Changes in manuscript:**
    The figures will be aligned.

6. *Page 13, line 1: " is A measure"*

    **Author's reply:**
    Suggestion followed.

    **Changes in manuscript:**
    Corrected.

---

## Author Comment (AC2) · 6 Jul 2020

**MS-No.:** os-2020-25
**Version:** Revision
**Title:** Multidecadal Preconditioning of the Maud Rise Polynya Region
**Author(s):** René M. van Westen and Henk A. Dijkstra
* * *
Interactive
comment

**Point-by-point reply to reviewer #2**

July 6, 2020

We thank the reviewer for his/her careful reading and for the useful comments on the manuscript.

Major comments:

1. *What is the Southern Ocean Mode? There are very few modeling studies that even consider it. Do the authors consider it to be a different mode from the Southern Annular Mode (SAM)? However, one of the definitions of SAM index is the difference between normalized zonally mean sea level pressure between 40S and 65S (Gong and Wang 1999). (SAMindex = P\*40S – P\*65S ) whose Atlantic sector encompasses the region used in the definition of SOM. The authors do not provide an explanation as to why the SOM index is an independent mode. The Southern Annular Mode could in fact influence the Temperature anomaly over 0-50W;35-50S. It would be very helpful if the authors can show that the SOM index is different from the SAM index. It is really not clear why this is a better metric or how this region behaves as an independent influence on the preconditioning of the polynyas over Maud Rise.*

   **Author's reply:**
   To provide more information on the relation between the SOM and SAM, we have

already performed additional analysis. First, we analysed the SOM using the stand-alone POP simulation which was first used to detect the SOM (Le Bars et al. 2016) using a Principal Component Analysis (PCA) of the upper ocean temperature fields in the Southern Ocean. A large-scale pattern of variability is found then with a period of about 40 – 50 years. In the stand-alone POP model, the associated PC displays the same multidecadal variability as the SOM index in Le Bars et al. (2016). This multidecadal variability is not related to any atmospheric variability since the atmospheric forcing is seasonally varying in the stand-alone POP.

Secondly, the same PCA (as above) is done for the CESM (fully coupled climate model). The first principal component displays a 25-year period and this period is significant against red noise in the CESM. The SOM index time series contains much more noise compared to the first principal component. The noise in the SOM index is likely related to atmospheric variability (which is absent in the stand-alone POP). Therefore, we do not use the SOM index in the revision anymore but the first principal component time series of the PCA.

Finally, we determined the SAM index in the stand-alone POP and in the CESM. The SAM index is constant in the stand-alone POP. For the CESM, the SAM index displays inter-annual variability over the 101-year period. In both the stand-alone POP and CESM results, we find no relation between the SOM and SAM.

**Changes in manuscript:**
We will include a new section in the revision: The Southern Ocean Mode. In this section, the PCA analysis is described and we motivate why the SOM is different from the SAM. In the revision, we will use the first principal component of the PCA to measure the SOM instead of the SOM index. All figures and results will be changed accordingly.

2. *The authors do not present a robust justification for using only the SOM and not looking at any of the important factors that play a role in preconditioning*

*such as the structure and strength of the Weddell gyre (Cheon et al. 2015) , the SAM index (Gordon et al. 2007), or the presence of stratified Taylor columns over Maud Rise (de Steur et al. 2007; Alverson and Owens 1996). Previous literature (Hirabara et al. 2012) suggests that the preconditioning and formation mechanism of these MRPs are related to several contributing factors that involve the interaction of large- and smallscale ocean/atmospheric circulation. The authors here do not provide a clear physical connection on how the SOM can in turn influence these other contributing factors for polynya formation. The lack of a physical mechanism clearly linking the SOM to the MRPs is a major weakness in this publication.*

**Author's reply:**
We have analysed the SAM index in CESM, but find no connection with the SOM (see also Major comment 1). We determined the zonal mass transport at 30°W as a measure for the Weddell gyre strength. The Weddell gyre strength also varies with the same 25-year period as the SOM. We found stratified Taylor column over Maud Rise in the CESM which are important for preconditioning the Maud Rise region. Changes in the surface salinity can also contribute to preconditioning. We already have analysed changes in the surface salinity but the surface is freshening prior to Polynya formation.

**Changes in manuscript:**
We will include the SAM index time series in the new section 'The Southern Ocean Mode', as well as the Weddell Gyre analysis. We will include the analysis of the surface salinity and the Taylor columns over Maud Rise. Preconditioning by surface salinity, Taylor columns and the SAM index will be discussed in section 'Preconditioning of the Polynya region'. The results and parts of the text will be changed accordingly.

3. *The loosely connected interpretation of figures 5,6 and 7 suggests that particles*

*released over the MRP box and backtracked for 8 years end up in specific regions in the Weddell gyre that are correlated to parts of the SOM region. However, correlation does not imply causation.*

**Author's reply:**
We have updated the lag-correlations patterns since we do not use the SOM index time series in the revision. The lag-correlation patterns overlap with the regions of where the particles end up, indicating that positive subsurface anomalies at that time propagate towards the Polynya region.

**Changes in manuscript:**
We will rewrite this paragraph of the manuscript and will also include lag-correlations patterns (cf. Major comment 4) to demonstrate the propagation of positive subsurface heat anomalies along the Weddell Gyre. The text, results and figures will be changed accordingly.

4. *If the subsurface OHC anomalies from the SOM region propagate along the Weddell gyre to the MRP box in ten years, one should be able to see a pattern in the correlation between SOM and H1000 in which the SOM leads H1000 in steps of 12-24 months (similar to figure 5d but in steps where SOM leads H1000 in 24, 48, 72, 96 months). It would help readers to see the intermediate steps of SOM leading H1000 by 12-24 months between figures 5c and 5d in which the propagation should be evident if such connections exist.*

**Author's reply:**
Suggestion followed.

**Changes in manuscript:**
We will include lag-correlation patterns in the revision. Note that we will not use the SOM index anymore for the lag-correlations. The results and figures will be changed accordingly.

Minor comments:

1. *The mapping projection in figure 5 is not consistent with figures 6-7. Please be consistent with the mapping projections.*

   **Author's reply:**
   Suggestion followed.

   **Changes in manuscript:**
   All the relevant figures are remapped and will be consistent.

2. *The authors mention 3 MRP events observed in 1973-1977,1994 and 2016-2017 to suggest a periodicity of 20 years between MRP events. They did not include the MRP event of 1980 (Comiso and Gordon 1987). The observations do not suggest a periodicity of 20 years. The polynyas of the late-70s were extremely large Weddell Sea Polynyas that begin over Maud Rise and spread westward into the Weddell Sea and have not been observed since.*

   **Author's reply:**
   Thanks for pointing this out.

   **Changes in manuscript:**
   The text will be rewritten accordingly.

---

## Referee Report (RR1)

Review of the manuscript:

Multidecadal Preconditioning of the Maud Rise Polynya Region

By Authors

René M. van Westen and Henk A. Dijkstra

Reviewer Decision: Acceptable after minor revision.

I am happy to see that the authors have addressed most of my mentioned concerns. However, the revisions bring into light some new concerns and I hope the authors can address them.

1) Has either SOM or SOM* ever been observed in nature? or any model other than CESM/POP? If not then explicitly state it and appropriately reduce the relevance throughout the manuscript. The authors on multiple occasions throughout the manuscript over sell this idea of SOM and SOM* but that is misleading since as far as we know this is a CESM/POP simulation artifact only captured in 4 CESM simulations so far. My main concern here is that the authors need to be upfront about the hypothetical nature of this study.

2) Lines 37-40 in Section 3: I'm really not convinced about your reasons for choosing different area for calculating SOM* index. The new area is (a) significantly smaller than the originally proposed area, and (b) is away from the region where EOF pattern has maxima/minima. Just because the correlation between SOM* index and Weddell Gyre strength is greater, that alone doesn't justify your choice.

3) The authors also need to make it clear that the fact that subsurface heat accumulation is useful in Weddell Sea Polynya formation in model studies, it has not been validated in any observational studies as a pre-requisite for Maud Rise Polynya formation.

4) Line 31 in Section 3: Why is there the phase difference between PC1 and SOM/SOM* index ? Is it significant ? Would it affect the proposed causal relationship with the Weddell Gyre strength?

5) Line 34 in Section 3: what is the correlation coefficient between PC1 and WG strength? for both models?

6) Lines 48-50 in Section 3: Despite the 60-month smoothing the percentage of variance explained is small implying a large part of regions variability is dominated by spatial noise. Correct?

7) Line 68 in Section 3: how long would a high-resolution CESM simulation need to be to have SOM? more than 100 years? Recent high-resolution model simulations that best reproduce the Maud Rise Polynya (Kurtakoti et al. 2018; Kaufman et al. 2020) show no evidence of SOM or SOM* in their ~130 years long CESM simulation. In fact, there is no periodicity in the subsurface heat accumulation in the Weddell Sea.

8) There is a line count reset in Section 3 after Fig. 3. Please fix that.

---

## Author Response (AR2)

**MS-No.:** os-2020-25

**Version:** Revision II

**Title:** Multidecadal Preconditioning of the Maud Rise Polynya Region

**Author(s):** René M. van Westen and Henk A. Dijkstra

**Point-by-point reply to reviewer #1**

October 6, 2020

We thank the reviewer for his/her careful reading and for the useful comments on the revised manuscript.

Major comments:

1. *CESM first EOF captures 4.1% of variance. This is a very small percentage. First it raises questions about whether it is significant or not. North (1992) (if I recall correctly) provides a method to evaluate the significance of EOFs. Are these others EOFs well separated from the first one?*

   *Secondly, I'm surprised that the authors gloss over this aspect. There is one remark that the CESM 1st EOF explains less variance than that of POP (which is already quite small). Are we looking at noise?*

   *This deserves more discussion. I expect that most readers will pause there. To reinforce the paper and its interest, this aspect would need to be addressed.*

   **Author's reply:**
   The variance captured by the first EOF is indeed quite small for both the POP and CESM. The low variance of the first EOF/PC is related to spatial noise in the ocean, the total number of grid cells (so size of the region) and the length of the time series. For example, determining the EOF over the smaller region of $50°W – 30°E × 40°S – 70°S$ increases the variance of the first EOF to 31.3% (POP) and 7.8% (CESM). Applying a 60-month running mean (as stated in the manuscript) reduces the noise in the time series and consequently increases the variance. In each

case, however, the same large-scale pattern of multidecadal variability emerges in PCA. Following North et al. (1982), we find that the first and second PC are well separated for both POP ($\lambda_1 = 16\%$, $\Delta\lambda_1 = 0.1\%$, $\lambda_2 = 8.3\%$) and CESM ($\lambda_1 = 4.1\%$, $\Delta\lambda_1 = 0.03\%$, $\lambda_2 = 2.4\%$).

To demonstrate that the first EOF is not significant against red noise (null hypothesis). we generated red-noise surrogate temperature fields with the same statistical properties as the pre-filtered temperature time series. We conducted EOF analyses using these surrogate fields and retaining the first EOF/PC (as is done for POP and CESM). We analysed 250 surrogate fields and the variance of the first PC is 3.8% and 0.9% for the POP and CESM at the 99%-Cl, respectively. The first EOF/PC of the 'real' data is 16% (POP) and 4.1% (CESM), both the POP and CESM exceed the variance of red noise and hence these EOFs are significantly different than red noise.

**Changes in manuscript:**
In the revision we discussed in more detail the variance of the first EOF for the POP and CESM. In addition, we included the results of the red-noise significant tests to show that the PCs/EOFs are significantly different from red noise.

2. *line 23, page 1: 'a negative freshwater flux' it doesn't mean much without being clear about the sign convention for the fluxes. Maybe use a more explicit term such 'net evaporation'*

   **Author's reply:**
   Suggestion followed.

   **Changes in manuscript:**
   We used the term 'net evaporation'.

3. *line 20-23, page 2: 'Analysis ,,, (Juling et al, 2018)'. Strange sentence. Responsible for what?*

   **Author's reply:**
   We agree with the reviewer.

   **Changes in manuscript:**
   These sentences are rewritten.

4. *line 19-20, page 5 : 'The SOM is a large-scale intrinsic ocean mode in the Southern Ocean (Le Bars et al., 2016) and was first found in a*

*stand-alone ocean simulation of the POP (the ocean component of the CESM).'*

*The first part of the sentence make it sound like the SOM is an established observed mode. Please rephrase to make it clear that has only been seen in models at this point (well one model really, POP).*

**Author's reply:**
Agreed.

**Changes in manuscript:**
We changed the text accordingly.

5. *line 6-7, page 12: 'The individual particles are hard to distinguish in Figure 7b, therefore we present the final distributions of the particles which are initially released at 200 m, 500 m and 1000 m depths in the Polynya region, are shown in Figure 8a, 8c and 8e, respectively.' Something strange, please rephrase.*

**Author's reply:**
Indeed, this does not make sense.

**Changes in manuscript:**
We changed the text accordingly.

6. *Line 11, page 17: 'and eventually they reach' remove they*

**Author's reply:**
Suggestion followed.

**Changes in manuscript:**
Corrected.

**MS-No.:** os-2020-25

**Version:** Revision II

**Title:** Multidecadal Preconditioning of the Maud Rise Polynya Region

**Author(s):** René M. van Westen and Henk A. Dijkstra

**Point-by-point reply to reviewer #2**

**October 6, 2020**

We thank the reviewer for his/her careful reading and for the useful comments on the revised manuscript.

Major comments:

1. *Has either SOM or SOM* ever been observed in nature? or any model other than CESM/POP? If not then explicitly state it and appropriately reduce the relevance throughout the manuscript. The authors on multiple occasions throughout the manuscript over sell this idea of SOM and SOM* but that is misleading since as far as we know this is a CESM/POP simulation artifact only captured in 4 CESM simulations so far. My main concern here is that the authors need to be upfront about the hypothetical nature of this study.*

   **Author's reply:**
   Indeed, the SOM variability emerges in long ($> 150$ years) simulations using strongly eddying oceans, of which there are only a few (basically only CESM and POP as far as we know). Whether the SOM is an artifact in these models will become clear as soon as simulations from similar other models become available.

   **Changes in manuscript:**
   In the revision we have emphasized that the SOM is only found in the CESM and POP.

2. *Lines 37–40 in Section 3: I'm really not convinced about your reasons for choosing different area for calculating SOM* index. The new area is (a) significantly smaller than the originally proposed area, and (b) is away from the region where EOF pattern has maxima/minima. Just*

*because the correlation between SOM\* index and Weddell Gyre strength is greater, that alone doesn't justify your choice.*

**Author's reply:**
We agree that the choice of the SOM\* region needs to be better motivated. We have tested the sensitivity of the results to the SOM\* region. Following the EOF pattern, we have extended it to 30°W – 20°E × 55°S – 40°S, which covers the maxima for both EOF patterns. This does not essentially change the results (which are in the revision). We didn't extend the SOM\* region down to 60°S (where the EOF pattern is positive for the CESM) since we want to exclude MRP variability in the SOM\* region. This new SOM\* index decreases the phase difference between the PC and the SOM\* index for the POP (46 months → 29 months) and CESM (36 months → 19 months).

**Changes in manuscript:**
All the SOM\* related results will be updated according to the choice of the new region (30°W – 20°E × 55°S – 40°S), which is now better motivated.

3. *The authors also need to make it clear that the fact that subsurface heat accumulation is useful in Weddell Sea Polynya formation in model studies, it has not been validated in any observational studies as a prerequisite for Maud Rise Polynya formation.*

**Author's reply:**
The reviewer is correct that prior to the 2016 – 2017 MRP there are no observations of long-term subsurface heat accumulation near Maud Rise (see for example Campbell et al. (2019)). However, Smedsrud et al. (2005) and Fahrbach et al. (2004, 2011) observe that the water mass near Maud Rise considerably warmed over longer periods of time prior to polynya formation.

**Changes in manuscript:**
We mention this issue now in the revised paper.

4. *Line 31 in Section 3: Why is there the phase difference between PC1 and SOM/SOM\* index ? Is it significant ? Would it affect the proposed causal relationship with the Weddell Gyre strength?*

**Author's reply:**
As demonstrated in the stand-alone POP, the variability induced by the SOM can be measured using the SOM index. The SOM index is almost in phase with the first PC and significant lag-correlations exists between the SOM index and the PC (65 months). One can change this phase difference between the first PC and the SOM index when altering the SOM region. For example, determining the SOM index over the region of 50°W – 0° × 70°S – 60°S (the EOF has a negative amplitude negative here), results in negative correlations between the first PC and SOM index. This change in phase difference does not affect the causal relationship with the Weddell Gyre strength.

**Changes in manuscript:**
We included the significances and phase differences of the lag-correlations in the revision.

5. *Line 34 in Section 3: what is the correlation coefficient between PC1 and WG strength? for both models?*

**Author's reply:**
The PC1 and WG strength in the POP are 180° out of phase with a significant correlation of $r = -0.90$. For the CESM we find a significant lag-correlation of $r = -0.62$ where the first PC leads by 93 months.

**Changes in manuscript:**
In the revision, we included the lag-correlation coefficient between the first PC and WG strength.

6. *Lines 48–50 in Section 3: Despite the 60-month smoothing the percentage of variance explained is small implying a large part of regions variability is dominated by spatial noise. Correct?*

**Author's reply:**
Determining the EOF over a smaller region (50°W – 30°E × 40°S – 70°S) increases the variance of the first EOF to 31.3% (POP) and 7.8% (CESM) for the non-smoothed time series. The variance increases further when the time series of the smaller region are smoothed through 60-month running mean, the variance is in this case 47.1% (POP) and 23.1% (CESM).

**Changes in manuscript:**
The PCA sensitivity is included in the revised manuscript.

7. *Line 68 in Section 3: how long would a high-resolution CESM simulation need to be to have SOM? more than 100 years? Recent high-resolution model simulations that best reproduce the Maud Rise Polynya (Kurtakoti et al. 2018; Kaufman et al. 2020) show no evidence of SOM or SOM\* in their 130 years long CESM simulation. In fact, there is no periodicity in the subsurface heat accumulation in the Weddell Sea.*

   **Author's reply:**
   The stand-alone POP was spun up for 75 years. In the first 50 model years (75 − 125), the magnitude of the SOM related multidecadal variability is much smaller compared to the remaining part of the simulation (model years 125 − 275). This suggests that long simulations are needed before climate simulations develop the SOM mechanism. The SOM related multidecadal variability is present in the CESM, the CESM was spun up for 150 years and run for another 150 years.

   **Changes in manuscript:**
   We indicated that long simulations are needed before the SOM variability can be detected in high-resolution models.

8. *There is a line count reset in Section 3 after Fig. 3. Please fix that.*

   **Author's reply:**
   Suggestion followed.

   **Changes in manuscript:**
   Corrected.

[revised manuscript text omitted]

---

## Author Response (AR3)

**MS-No.:** os-2020-25

**Version:** Revision III

**Title:** Multidecadal Preconditioning of the Maud Rise Polynya Region

**Author(s):** René M. van Westen and Henk A. Dijkstra

**Point-by-point reply to Editor**

October 16, 2020

We thank the editor (Mario Hoppema) for the careful reading and for the useful comments on the revised manuscript.

1. *P1, L19: 'Convection below the region of the MRP' I think it is not clear what is meant with this. Please modify the text.*

   **Author's reply:**
   We agree that this is not very clear.

   **Changes in manuscript:**
   convection below the region of the MRP → in the vicinity of Maud Rise.

2. *P3, L9–10 'start equilibrating after about 150 years.' What is the criterion for starting to equilibrate? And what is the criterion for accepting the increase in the radiative imbalance?*

   **Author's reply:**
   What we meant was that after model year 150, the transient adjustment to the forcing becomes much smaller compared to the beginning of the simulation.

   **Changes in manuscript:**
   We have rewritten this paragraph.

3. *P3, L9–10 'The radiative imbalance is only slightly positive over the last 100 years of the simulation.' I think you mean here that it is slightly increasing, right? It is positive all the time.*

**Author's reply:**
No, it is not increasing. By mentioning the positive radiative imbalance, we want to indicate here that the CESM is not yet in radiative equilibrium .

**Changes in manuscript:**
No changes.

4. *P3, L11-12 'relatively small trends (compared to the first part of the simulation) occur over the last 100 years which can easily be removed through a quadratic detrending.' The figure shows that the trends between years 80 and 150 are exactly as large as those in the last 100 years.*

   **Author's reply:**
   Correct, the OHC trends are similar when determined over model years 80 – 150 and 150 – 250. However, this is not the true for the global mean surface temperature and radiative imbalance.

   **Changes in manuscript:**
   compared to the first part of the simulation $\rightarrow$ compared to the first 100 model years of the simulation

5. *P3, L17 'reasonably agrees with observations'. How is reasonably defined here? Please be more quantitative.*

   **Author's reply:**
   We determined the Antarctic sea-ice area for the CESM and observations. The modelled and observed September sea-ice area varies between 15.8 – 18.4 (101 model years) and 18.8 – 21.3 (19 years) million km$^2$, respectively.

   **Changes in manuscript:**
   We included the modelled and observed sea-ice areas in the revision.

6. *P5, L20 e.g. does not fit here, probably better: i.e.*

   **Author's reply:**
   Agreed.

   **Changes in manuscript:**
   Corrected.

7. *P8, L7 ... a SOM period of about 40 – 50 years (add: SOM)*

   **Author's reply:**
   Agreed.

   **Changes in manuscript:**
   Corrected.

8. *P8, L7 Please be more quantitative about 'similar'. What is similar here?*

   **Author's reply:**
   Here we refer to a similar SOM period.

   **Changes in manuscript:**
   similar → a similar period.

9. *P8, L22 delete: below (superfluous)*

   **Author's reply:**
   Agreed.

   **Changes in manuscript:**
   Corrected.

10. *P8, L27 Change the second 'sea-ice fractions' to values: The September sea-ice fraction (blue curve in Figure 4a) shows relatively low values with respect to ...*

    **Author's reply:**
    Agreed.

    **Changes in manuscript:**
    Corrected.

11. *P10, L10 As demonstrated in section 4.1, both ... (use section number)*

    **Author's reply:**
    Agreed.

    **Changes in manuscript:**
    Corrected.

12. *P10, L16 ... by 80 months. (why give the imprecise number)*

**Author's reply:**
The data analysed consists of monthly averages and lag-correlation analysis indicates an optimum lag at 80 months.

**Changes in manuscript:**
No changes.

13. *P11, L8 OceanParcels*

    **Author's reply:**
    Agreed.

    **Changes in manuscript:**
    Corrected.

14. *P13, L12-14 This sentence is not clear to me. Please rephrase and make clear.*

    **Author's reply:**
    The sentence is indeed confusing.

    **Changes in manuscript:**
    The sentence is rewritten.

15. *P16, L7 I suggest to emphasize the real years here as the transition from model years to real years comes unexpected to the reader: Note that observations show that prior to the MRP of the year 2016-2017, no ...*

    **Author's reply:**
    We agree that this is unexpected in the manuscript.

    **Changes in manuscript:**
    This is rephrased in the revision.

16. *P16, L14 'where 1 Sv Psu is about $10^9$ g of salt per second' The salinity can be expressed on the practical salinity scale, where salinity is dimensionless (psu is sometimes used but is not an existing unit). How is g salt calculated from this?*

    **Author's reply:**
    Psu is indeed dimensionless and it is defined as the amount of salt dissolved in 1 kg of sea water. For example, 35 Psu is 35 gram dissolved in 1 kg of sea water. 1 Sv Psu $= 10^6$ m$^3$ s$^{-1}$ Psu. As 1 m$^3$ of sea water

weights about $10^3$ kg, thus $10^9$ kg s$^{-1}$ g kg$^{-1}$, 1 Sv Psu is about $10^9$ g of salt per second.

**Changes in manuscript:**
No changes.

17. *P16, L22 ... over the Polynya region which are also ... (which instead of and)*

    **Author's reply:**
    Agreed.

    **Changes in manuscript:**
    Corrected.

18. *P16, L32-34 'The potential density at 900 m depth for each September is shown in Figure 10b and also shows Taylor columns over Maud Rise during other years.' This sentence is grammatically not correct.*

    **Author's reply:**
    We agree.

    **Changes in manuscript:**
    Sentence is rewritten.

19. *P16, L34 'stratification at 900 m depth' Stratification at a certain depth does not exist, right? It is always along a depth range. Please use correct terminology.*

    **Author's reply:**
    Correct, this is not the stratification but the potential density.

    **Changes in manuscript:**
    stratification $\rightarrow$ potential density

20. *P17, L9 delete: In this paper, (superfluous)*

    **Author's reply:**
    Agreed.

    **Changes in manuscript:**
    Corrected.

21. *P17, L9 ... from a 250 years control simulation ... (2.5 centuries is not multi-century; multi = many)*

**Author's reply:**
Agreed.

**Changes in manuscript:**
Corrected.

22. *P17, L12 'broadly in agreement' What does this mean. Please explain where the agreement lies and where not.*

    **Author's reply:**
    Deep convection followed by MRP formation is also found in other model studies. The difference between the CESM and other (low-resolution) models is that deep convection occurs rather randomly in most model studies.

    **Changes in manuscript:**
    We now discuss the differences in the re-occurrence period of the MRP between the CESM and other models.

23. *P20, L12 Please add country*

    **Author's reply:**
    Agreed.

    **Changes in manuscript:**
    Corrected.

24. *P20, L11-13 Please indicate what this is: a technical report, etc. and access date.*

    **Author's reply:**
    Agreed.

    **Changes in manuscript:**
    Corrected.

25. *P20, L23 Add pages*

    **Author's reply:**
    Agreed.

    **Changes in manuscript:**
    Corrected.

26. *P20, L25 Reference is incomplete*

   **Author's reply:**
   Agreed.

   **Changes in manuscript:**
   Corrected.

27. *Fig 9 panel (a): Near the color scale is the term heat advection. This is misleading, as with advection one would expect a different unit than TW. If this concerns heat indeed (and thus TW), the term should be changed: advective heat Similar to panel 9 (b) and panels 9 (c) and 9 (d)*

   **Author's reply:**
   Agreed.

   **Changes in manuscript:**
   Corrected in labels of the figures.

28. *In Figure 1 it is shown that the model is not stable after 250 years, especially for the OHC. Still, you draw conclusions about the heat influence in the MRP region. Please explain why this is not a problem.*

   **Author's reply:**
   The integrated OHC time series over the Polynya region (as in Figure 4a) does not display strong drifts compared to the time series as in Figure 1b. Note that the red curve in Figure 1b is the upper 1000 m integrated OHC over the Southern Hemisphere and not representative for the much smaller Polynya region.

   **Changes in manuscript:**
   In the revision, we added that the drift in the integrated OHC over the Polynya region is smaller compared to the ones in Figure 1b.

29. *P5, L19 '(model years 175 − 275).' I am a little confused. In section 2 the authors wrote that only the model years 1-250 would be used.*

   **Author's reply:**
   In section 3, two simulations are analysed: 1) the stand-alone POP and 2) the CESM. For both simulations we analyse the ocean component (i.e. the POP). The main difference between the simulations is that atmospheric and sea-ice related processes are not dynamically resolved

in the stand-alone POP. For the stand-alone POP model years 175 – 275 are analysed. For the CESM, the first 250 model years are analysed.

**Changes in manuscript:**
We clarified this in the revision.

[revised manuscript text omitted]